# Fine-Grained Human Feedback Gives Better Rewards for Language Model Training

**Zeqiu Wu**[1][*]    **Yushi Hu**[1][*]    **Weijia Shi**[1]    **Nouha Dziri**[2]    **Alane Suhr**[3]
**Prithviraj Ammanabrolu**[45]    **Noah A. Smith**[12]    **Mari Ostendorf**[1]    **Hannaneh Hajishirzi**[12]
[1]University of Washington    [2]Allen Institute for Artificial Intelligence
[3]University of California, Berkeley    [4]University of California, San Diego    [5]MosaicML

## Abstract

Language models (LMs) often exhibit undesirable text generation behaviors, including generating false, toxic, or irrelevant outputs. Reinforcement learning from human feedback (RLHF)—where human preference judgments on LM outputs are transformed into a learning signal—has recently shown promise in addressing these issues. However, such holistic feedback conveys limited information on long text outputs; it does not indicate which aspects of the outputs influenced user preference; e.g., which parts contain what type(s) of errors. In this paper, we use fine-grained human feedback (e.g., which sentence is false, which sub-sentence is irrelevant) as an explicit training signal. We introduce FINE-GRAINED RLHF, a framework that enables training and learning from reward functions that are fine-grained in two respects: (1) density, providing a reward after every segment (e.g., a sentence) is generated; and (2) incorporating multiple reward models associated with different feedback types (e.g., factual incorrectness, irrelevance, and information incompleteness). We conduct experiments on detoxification and long-form question answering to illustrate how learning with such reward functions leads to improved performance, supported by both automatic and human evaluation. Additionally, we show that LM behaviors can be customized using different combinations of fine-grained reward models. We release all data, collected human feedback, and codes at https://FineGrainedRLHF.github.io.

## 1   Introduction

State-of-the-art AI is built on pre-trained language models that are then trained through interaction with humans [29, 28, 9], with a combination of supervised learning and reinforcement learning. Incorporating human feedback into the process of language model (LM) training has been shown as effective to reduce false, toxic and other undesired model generation outputs [29, 3, 2, 33, 10]. Many of these studies adopt reinforcement learning from human feedback (RLHF) [46], a framework that converts human feedback into an effective LM training signal to reach these goals. Specifically, humans are presented with two or more outputs and asked to select one or rank them, and this signal is then used to train a reward model, which computes a single scalar reward for each LM-generated sequence. The LM is then trained with RL to optimize the reward it receives (from the reward model).

Such a reward provides a relatively sparse training signal, especially for tasks that require the generation of long-form text—making RLHF in such domains unreliable [33]. Furthermore, previous research [8, 18, 11, 45, 25, 43] into automated evaluation of generated text shows that it can be challenging for human annotators to reliably compare the overall quality of two or more model outputs when the outputs contain a mixture of diverse undesired behaviors. They demonstrate how categorizing and localizing model errors (i.e., fine-grained evaluation) provides explicit insights about which part of the model output has what type of problem. We thus ask the question: how can we improve rewards for LM training via RLHF by using more *fine-grained human feedback*?

---

[*]Equal contribution. Correspondence to <Zeqiu Wu: zeqiuwu1@uw.edu>, <Yushi Hu: yushihu@uw.edu>

37th Conference on Neural Information Processing Systems (NeurIPS 2023).

Figure 1: Comparison of **(a) RL with human preference** and **(b) our FINE-GRAINED RLHF** on long-form QA. Different from (a), which collects human preferences on the overall quality of LM outputs, we ask annotators to mark which part of an output contains what type(s) of errors. We train a fine-grained reward model for each type of error and optimize LM against these reward models. In this example, we provide a relevance reward and a factuality reward after each sentence is generated. There is also a holistic information completeness reward after the whole text is generated.

In this paper, we propose that humans give fine-grained feedback to LM output, associating *categories* of undesired behavior (e.g., false or irrelevant generations) and a text span at a *density* (e.g., sentence or sub-sentence-level). To enable LMs to learn from such fine-grained feedback, we introduce the FINE-GRAINED RLHF framework. As shown in Figure 1, we first use collected human feedback to train fine-grained reward models such that each of them focuses on one *category* and provides rewards at the *density* associated with that category. We then integrate these reward models into Proximal Policy Optimization (PPO) [37], a commonly used RL algorithm for training LMs with preference-based human feedback (§2).

We conduct experiments on two language generation tasks—detoxification [12] (§3) and long-form question answering (QA) [39] (§4). For detoxification, toxicity is the only error category and we explore learning with a dense reward. We adopt PERSPECTIVE [1], a widely used language toxicity detection model trained on millions of human annotations, as our reward model. We use it to calculate a fine-grained reward after the generation of every sentence. Our experimental results show the efficacy and data efficiency of training models with dense reward compared to a holistic sequence-level reward, supported by automatic evaluation results.

With experiments on long-form QA, we aim to examine training models with fine-grained rewards at the two granularity dimensions (density and error category), for which we construct a long-form QA dataset, QA-FEEDBACK, along with our collected human feedback. We carefully develop a pipeline to collect fine-grained human feedback on three error categories at different density levels: i) irrelevance, repetition, or incoherence (sub-sentence), ii) incorrect or unverifiable facts (sentence), and iii) incomplete information (whole sequence; see Figure 1). Our experimental results show improved results in each error category by learning with such fine-grained feedback, supported by both automatic and human evaluation results. In a scenario with multiple reward models representing different error types, we also show FINE-GRAINED RLHF allows us to combine reward models with different weights and thus control the model training process towards a customized combination of desired behaviors.

## 2 FINE-GRAINED RLHF

We introduce FINE-GRAINED RLHF, a framework that enables us to train fine-grained reward functions for generation outputs across different feedback types. We first define the RL environment and learning algorithm. Then we define the fine-grained reward models and describe how to incorporate the fine-grained reward model(s) into an RL algorithm, in contrast to previous RLHF studies that only consider a single reward.

**Environment: language generation as a MDP.** We focus on language generation tasks. For each task, we are given a set of task input prompts $D = \{x^n\}_{n=1}^N$. We follow [33] to define language generation as a Markov Decision Process (MDP) $\langle \mathcal{S}, \mathcal{A}, \mathcal{R}, P, \gamma, T_{max} \rangle$ with a finite vocabulary $\mathcal{V}$. Each MDP episode starts with a sampled prompt $x = (x_1, x_2, \ldots, x_l)$ with $x_i \in \mathcal{V}$, and ends when the current time step exceeds $T_{max}$ or an end of sequence token is generated. $\mathcal{S}$ is the state space and $s_0 = (x_1, x_2, \ldots, x_l) \in \mathcal{S}$ is the initial state. An action in the environment $a_t \in \mathcal{A}$ is a generated token (by the policy LM model $P_\theta$) at time $t$ from $\mathcal{V}$ ($a_0$ is the begin sequence token). The transition function $P : \mathcal{S} \times \mathcal{A} \to \Delta\mathcal{S}$ appends $a_t$ at the end of the state $s_t = (x_1, x_2, \ldots, x_l, a_0, a_1, \ldots, a_{t-1})$. This process continues until the end time step $T \leq T_{max}$ is reached, which gives a generated sequence $y = (a_1, \ldots, a_T)$. A reward function $\mathcal{R} : \mathcal{S} \times \mathcal{A} \to \mathbb{R}$, which comes from the reward model(s) in FINE-GRAINED RLHF, provides dense rewards before and when $T$ is reached. $P_\theta$ can be initialized with a pre-trained language model, and sometimes also with supervised fine-tuning on task-specific demonstrations. The reward function is defined later.

**Learning algorithm: proximal policy optimization (PPO).** PPO [37] is an actor-critic RL algorithm that is widely used in previous RLHF work to optimize the policy model against a reward model of human feedback. It uses a value model $V_\psi(s_t)$ to estimate the value of state $s_t$, and optimizes the policy model with a PPO clipped surrogate training objective. The advantage $A_t$ at timestep $t$ is estimated by a generalized advantage estimation function [36]: $A_t = \sum_{t'=t}^{T} (\gamma\lambda)^{t'-t}(r_{t'} + \gamma V_\psi(s_{t'+1}) - V_\psi(s_{t'}))$, with $\gamma$ as a hyperparameter and $\lambda$ as the discounting factor for rewards. $r_t$ is the reward assigned to $a_t$, which in our case is acquired using one or multiple learned reward models. The value model $V_\psi(s_t)$ is optimized with an expected squared-error loss with the value target as $V^{\text{targ}}(s_t) = \sum_{t'=t}^{T-1} \gamma^{t'-t} r_{t'} + \gamma^{T-t} V_{\psi_{\text{old}}}(s_T)$, where $V_{\psi_{\text{old}}}$ is the lagging value model. Finally, PPO is trained to optimize both policy ($P_\theta$) and value ($V_\psi$) models with their respective objectives. No reward model is being optimized during PPO training. See Appendix B for more details.

**Fine-grained reward models.** Previous RLHF work adopts a holistic reward model $R_\phi$ that maps input prompt $x$ and generated output $y$ to a single scalar reward representing its overall quality (Figure 1(a)). This single scalar reward is only assigned to the final token in the generated sequence, $a_T$. Formally, $r_t = R_\phi(x, y)$ if $t = T$ and 0 otherwise.

In contrast, we consider a reward function that is derived from one or multiple *fine-grained* reward models that (1) provide rewards densely (i.e., for subsequences of the generated output), and (2) compute rewards on distinct categories of undesired behaviors (e.g., false or repetitive generation), where each category is associated with an individual reward model.

For a fine-grained reward model $R_{\phi_k}$ that gives feedback on error category $C_k$, we first segment $y$ into $L_k$ segments $(y_1^k, y_2^k, \ldots, y_{L_k}^k)$ corresponding to the density (e.g., sentence-level) of $R_{\phi_k}$, where each segment $y_j^k$ ends at timestep $T_j^k$. $R_{\phi_k}$ outputs a reward $R_{\phi_k}(x, y, j)$ for each segment $y_j^k$ given $x$ and $y$ as the input, which is assigned to the final token in $y_j^k$. Additionally, to ensure the fluency of generated outputs, we follow [41] to add an approximate KL divergence penalty to each token $a_t$ with a weight $\beta$, that is not backpropagated through during training. Formally, assuming that we have $K$ fine-grained reward models that represent different error categories, we will have a combined reward function for each token $a_t$ as:

$$r_t = \sum_{k=1}^{K} \sum_{j=1}^{L_k} \left( \mathbb{1}(t = T_j^k) \, w_k \, R_{\phi_k}(x, y, j) \right) - \beta \log \frac{P_\theta(a_t \mid s_t)}{P_{\theta_{\text{init}}}(a_t \mid s_t)} \tag{1}$$

where $w_k \in \mathbb{R}$ is a weight assigned to reward model $R_{\phi_k}$. Then we follow the same PPO training algorithm to optimize the policy model. We discuss how we define and train fine-grained reward models for the detoxification and long-form QA task in our experiments in § 3 and § 4 respectively.

# 3 Task 1: Detoxification

The task of detoxification aims to reduce the toxicity in the model generation $y$ when given a prompt $x$. Toxicity is the only undesired behavior in this task, and we aim to explore learning with a dense reward in comparison to a *single* holistic reward. We conduct our experiments on REALTOXICITYPROMPTS, a dataset of 100K sentence-level prompts derived from the web that are known to easily elicit problematic generations in GPT-2 [31]. Using a dense sentence-level fine-grained reward, we demonstrate that **our fine-grained reward exhibits greater sample efficiency compared to a holistic reward**, achieving lower toxicity with fewer training steps while maintaining better fluency (§3.1).

**Holistic reward for (non-)Toxicity.** We use the PERSPECTIVE API [1] as our reward model, which is widely used for language toxicity detection and is trained with millions of examples gathered from several online platforms and annotated by human annotators for toxicity. That means we use an off-policy reward model that is not trained on outputs from $P_{\theta_{init}}$. The API outputs a score between 0 (non-toxic) and 1 (toxic). Given the entire model output $y$, the holistic reward for RL is $1-$PERSPECTIVE$(y)$.

**Sentence-level (fine-grained) reward for (non-)Toxicity.** To calculate the *fine-grained reward*, we query the API after the model generates each sentence instead of generating the full sequence. For each generated sentence $y_j$, we assign PERSPECTIVE$([y_1, \ldots, y_{j-1}])$ - PERSPECTIVE$([y_1, \ldots, y_j])$ as the sentence reward (i.e., how much toxicity is changed from generating $y_j$). Since there is only one error category, we omit the category superscript, using $y_j$ to denote the $j^{th}$ segment (e.g., sentence) in $y$.

## 3.1 Experiments

**Implementation details.** We follow previous work [17, 21] and use GPT-2 large model as the initial policy model $P_{\theta_{init}}$. During both the exploration stage in RL training and inference, we use nucleus sampling decoding with $p = 0.9$ and temperature = 1.0. The generation length limit is set to 48. The value model used during RL training is initialized with GPT-2-base due to GPU memory constraint. We report RL training parameters in Appendix B. All scores are averaged over 3 independent runs.

**Compared systems and evaluation.** We report the performance of **FINE-GRAINED RLHF**, RLHF with holistic reward (**Hol. RLHF**), and the state-of-the-art controlled generation approaches **GeDi** [17] and **DEXPERTS** [21]. We follow previous work [17, 21] to report the toxicity score calculated on each full generation sequence from the PERPLEXITY API, as well as other commonly used metrics for REALTOXICITYPROMPTS, including n-gram diversity and GPT-2 XL perplexity (PPL) as a proxy for fluency. The lower the perplexity, the more fluent the generated text. The toxicity score is reported as the *maximum* score among 4 sampled model outputs, averaged over all test input prompts. Other metrics are reported as the *average* score of the same 4 samples.

| | Toxicity | Fluency | Diversity | |
| --- | --- | --- | --- | --- |
| | avg max ($\downarrow$) | PPL ($\downarrow$) | dist-2 ($\uparrow$) | dist-3 ($\uparrow$) |
| GPT-2 | 0.192 | 9.58 | 0.947 | 0.931 |
| *Controlled Generation* | | | | |
| GeDi | 0.154 | 24.78 | 0.938 | **0.938** |
| DEXPERTS | 0.136 | 22.83 | 0.932 | 0.922 |
| Hol. RLHF | 0.130 | 11.75 | 0.943 | 0.926 |
| **F.G. RLHF** | **0.081** | **9.77** | **0.949** | 0.932 |

Table 1: Results on the REALTOXICITYPROMPTS test set.

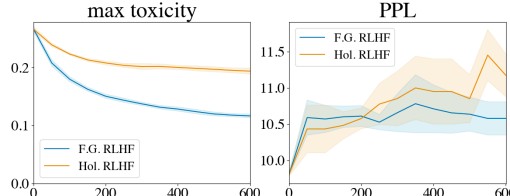

Figure 2: Curves of toxicity and perplexity on the dev set vs. training steps.

**Main results.** Table 1 shows the experimental results on the REALTOXICITYPROMPTS test set. FINE-GRAINED RLHF with sentence-level fine-grained reward attains the lowest toxicity and perplexity among all methods, while maintaining a similar level of diversity.

**Sample efficiency analysis.** Figure 2 shows the max toxicity and average perplexity on the development set during training. FINE-GRAINED RLHF has the toxicity drop much faster while keeping a low-level perplexity. This shows that learning from denser fine-grained reward is more sample efficient than holistic reward. One explanation is that fine-grained reward locates where the toxic

content is, which is a stronger training signal compared with a scalar reward for the whole text. The cost is that we have to query the reward model more times per example.

# 4 Task 2: Long-Form Question Answering (QA)

Long-form QA requires an LM to generate a textual response to a question with a comprehensive answer and explanation. To examine learning with fine-grained rewards at the two granularity dimensions (error category and density), we collect QA-FEEDBACK (§4.1), a long-form QA dataset annotated with human feedback on LM-generated responses. We define three error categories at different density levels and train a reward model for each (§4.2). We describe the experimental setup in §4.3. Both human and automatic evaluation show that FINE-GRAINED RLHF outperforms preference-based RLHF and supervised fine-tuning models on all error categories (§4.4). We then show that adjusting the weights of fine-grained reward models during RL training leads to distinct behaviors in LM generation, allowing us to customize the LM for users with different needs (§4.5). Finally, we conduct an in-depth analysis of the fine-grained reward models, revealing that they compete against each other, and provide an analysis of their impact on the resulting policy model.

## 4.1 QA-FEEDBACK: Long Form QA with Human Feedback

QA-FEEDBACK is based on ASQA [39], a dataset that focuses on answering ambiguous factoid questions [26] in an open-domain setting. We use their provided oracle knowledge contexts to reformulate the task into a reading comprehension setting: given the input $x$ that contains a question $q$ and a set of knowledge passages $P = \{p_1, \ldots, p_{|P|}\}$, generate a long-form response $y$. On average, there are roughly 65 words in each gold response. Since ASQA does not release the test set, we create our own train/development/test data split from the original train and development sets. We name our newly constructed data, along with collected human feedback (discussed next), QA-FEEDBACK. Overall, we have 3,853 training, 500 development, and 948 test examples (details in Appendix C).

**Initial policy and fine-grained human feedback.** Before collecting human feedback, we follow [33] to initialize the policy model with supervised fine-tuning on a small set of examples. Specifically, we use 1K training examples to supervise fine-tuning of T5-large (the original baseline for ASQA) [32] to get $P_{\theta_{init}}$. We name this initial policy model **SFT**. We then sample outputs from SFT for the remaining training and development examples and collect *fine-grained* human feedback in three error categories— $C_1$: irrelevance, repetition, or incoherence ; $C_2$: incorrect or unverifiable facts based on knowledge passages; and $C_3$: incomplete information . The collected feedback instances are then used as the training and development examples for training reward models. For each task prompt $x$, we only collect fine-grained feedback for *one* model output. Our data collection has IRB approval and is deemed exempt.

We instruct workers to identify any error in each model output $y = (a_1, \ldots, a_T)$, marking the span of text associated with each identified error type. Formally, we define the set of user-annotated feedback for a task prompt $x$ and model output $y$ as $\mathcal{F} = \{f_i\}$ where each $f_i = \langle c_i, b_i, e_i \rangle$ represents the user-identified span $(a_{b_i}, \ldots, a_{e_i})$ of the error category $C_{c_i}$, where $c_i \in \{1, 2, 3\}$. Importantly, we impose three restrictions in the annotation: (1) error spans of category $C_1$ or $C_2$ should not overlap with each other; (2) only spans that do not have error $C_1$ need to be assessed as containing error $C_2$ or not; (3) $C_3$ can only apply to whole output sequences. Additionally, we ask workers to mark passage sentences that contain missing information if a $C_3$ error is annotated. We also ask workers to rewrite $y$ into a corrected version $y'$ that addresses all annotated feedback $\mathcal{F}$. Details about the feedback collection interface, instructions, and quality control are in Appendix C.

To analyze human-human agreement, a subset of 300 examples receive annotations from two distinct workers. We observe that while exact agreement in error span boundaries is low, workers achieve reasonably high agreement on whether a sub-sentence contains $C_1$ and whether a sentence contains $C_2$.[2] Therefore, we decide to have the density for error type $C_1$, $C_2$, and $C_3$ as sub-sentence, sentence and full sequence. We provide more data analysis including human agreement in Appendix C.

**Preference-based human feedback.** For comparison purposes, we follow [29] to separately collect pairwise *human preferences* from the same group of workers. We sample 4 model outputs for each prompt $x$, which gives 6 pairs of model outputs. We ask the workers to indicate pairwise preferences

---

[2]We use spaCy [15] to segment generated model outputs into sentences. We then split sentences into sub-sentences using a comma or semicolon.

(ties are allowed) based on all errors they can find in each model output. They are not asked to explicitly annotate these errors.

**Annotation details.** On average, both annotation tasks of fine-grained and preference feedback for one question take a worker about 6 minutes to finish. In contrast, [39] report that they spend about 15 minutes to label a human-written response for each question, which is much more time-consuming than our feedback annotation. On average, we pay $1.65 per example for both tasks, leading to $16.50 hourly pay for our workers. We include details of the pay structure in Appendix C. We observe that human annotators can reach a higher agreement in each aspect of fine-grained feedback compared to pairwise comparisons because the feedback definitions are more concrete.

## 4.2 Fine-Grained Reward Models

We train three separate reward models $R_{\phi_1}$, $R_{\phi_2}$, and $R_{\phi_3}$ for $C_1$, $C_2$, and $C_3$ error categories respectively with a density of sub-sentence, sentence, and full sequence, respectively. Since reward models provide scalar reward scores and do not perform generation, we use the encoder-only Longformer-base [4] as our backbone model to handle long input sequences (more details of each reward model are in Appendix D).

$C_1$**: Irrelevance, repetition, or incoherence.** $R_{\phi_1}$ targets to predict whether each sub-sentence in $y$ contains a $C_1$ type error. We denote $y = (y_1^1, \ldots, y_{L_1}^1)$, where $y_j^1$ is the $j$th segment at $R_{\phi_1}$'s density (i.e., sub-sentence), with $L_1$ segments in total. We add a 2-class token-level classification layer (a single feed-forward layer) on the top of the Longformer encoder. The model input has the format of "question: $q$ answer: [sep] $y_1^1$ [sep] $y_2^1$ ...", and we take the classification output at each [sep] token to indicate whether the following $y_j^1$ contains a $C_1$ error. We do not add passages in the model input because, intuitively, the detection of $C_1$ errors does not depend on them. To train $R_{\phi_1}$, we apply a token-level classification loss to each [sep] token before $y_j^1$, where its gold label $g_j$ is "has error" if there is a $f_i \in \mathcal{F}$ that has $(a_{b_i}, \ldots, a_{e_i})$ overlapped with $y_j^1$ and $c_i = 1$, and "no error" otherwise. When $R_{\phi_1}$ provides a reward during RL training as in Eq. 1, we read a reward $R_{\phi_1}(x, y, j)$ for every $y_j^1$ given $x$ and $y$. We define $R_{\phi_1}(x, y, j) = +1$ if $R_{\phi_1}$ predicts "no error" for $y_j^1$ and $-1$ otherwise.

$C_2$**: Incorrect or unverifiable facts.** $R_{\phi_2}$ is developed for detecting a $C_2$ error at the sentence level in a similar way. The model input has the format of "question: $q$ context: $p_1$ $p_2$ ... answer: [sep] $y_1^2$ [sep] $y_2^2$ ...", where $p$'s denotes the grounding passages and $y_j^2$ represents the $j$th sentence. We train $R_{\phi_2}$ similarly to $R_{\phi_1}$, with one exception: as we instruct the workers not to annotate a $C_2$ error for a span that is already labeled as containing a $C_1$ error, we do not calculate loss on sentences that are labeled as containing $C_1$ but not $C_2$ during $R_{\phi_2}$ training.

$C_3$**: Incomplete information.** $R_{\phi_3}$ is trained to measure the information completeness of $y$, at the full sequence level. Motivated by [19], $R_{\phi_3}$ predicts a single scalar reward and is trained with a pairwise comparison loss [29]:

$$\mathcal{L}_r(\phi) = -\mathbb{E}_{(x, \bar{y}_p, \bar{y}_l) \sim D_p} \Big[ \log \big( \sigma(R_{\phi_3}(x, \bar{y}_p) - R_{\phi_3}(x, \bar{y}_l)) \big) \Big] \tag{2}$$

where $R_{\phi_3}(x, y)$ is the scalar output of the reward model for input $x$ and output $y$; $\bar{y}_p$ and $\bar{y}_l$ are sampled from the same input $x$, and $\bar{y}_p$ has less missed information compared with $\bar{y}_l$; $D_p$ contains the pairwise comparisons bootstraped from human feedback on $C_3$ errors (see details in Appendix D).

**Preference-based reward model.** The preference-based reward model is trained in a similar way to $R_{\phi_3}$, with $\bar{y}_p$ representing the human preferred response against $\bar{y}_l$ in the loss function Eq. 2. It outputs a scalar score for the given $x$ and $y$ that represents the overall response quality.

## 4.3 Experimental Setup

**Compared systems.** We compare our proposed method, FINE-GRAINED RLHF with the initial T5 policy model trained with 1K examples (**SFT**) and RLHF with holistic preference-based rewards (**Preference RLHF**). The reward models used in RLHF experiments are trained on 2.8K examples with annotated feedback (but no gold human response). For analysis, we also use the human gold responses of all training examples to finetune a fully supervised T5 model (**SFT-Full**). Notice that SFT-Full requires much higher annotation cost because it takes longer (15 minutes per example [39]) for annotators to draft long-form responses.

**Implementation details.** Our policy model is based on T5-large [32] and is supervised finetuned on 1K training examples, as explained in §4. During RL exploration, we use top-k ($k = 20$) sampling decoding with temperature = 0.7, which is set based on previous RLHF work [33]. The value model used during RL training is initialized with T5-base due to GPU memory constraint. The reward model weights we used in FINE-GRAINED RLHF are $w_1 = 0.3, w_2 = 0.5, w_3 = 0.3$, unless otherwise specified. Although we use three reward models during RL training, we only observe very small relative additional cost (roughly 1% training time) compared to preference RLHF. During inference, we use greedy decoding to generate responses. We report more details including RL training parameters in Appendix B. All scores reported are averaged over 3 independent runs.

**Evaluation.** We conduct both human and automatic evaluation. Human evaluation is run on 200 randomly sampled test set examples of QA-FEEDBACK to compare *Fine-Grained RLHF* with all baselines. Each model output is sampled from inference results of 3 training runs. We use the same protocol of feedback collection to have the same set of workers annotate spans in each model output that contain (1) irrelevance, repetition, or incoherence error **(rel.)** and (2) incorrect or unverifiable facts **(fact.)** . They are also asked to compare the information completeness **(comp.)** for each output pair. To report evaluation scores for *rel.* and *fact.* error spans, we first map them to their corresponding error type density (sub-sentence and sentence). Then we report the error rate for each error type, measured as the percentage of sub-sentences that contains this type of error. Since spans with *rel.* error are not checked for *fact.* error (discussed in §4.1), we exclude sub-sentences with only *rel.* error when report the error rate of *fact.* error. For automatic evaluation, we report RougeLSum [20] as used for the original ASQA data, as well as the score from each fine-grained reward model ($R_{\phi_1}$, $R_{\phi_2}$, and $R_{\phi_3}$). Specifically, we report the percentage of all sub-sentences (or sentences) in the test set predicted as "no error" by $R_{\phi_1}$ (or $R_{\phi_2}$). For $R_{\phi_3}$, we report the averaged output score for all test examples.

## 4.4 Main Results

Figure 3 shows the human evaluation results for *rel.* and *fact.* error types. Table 2 shows the human pairwise comparison results for information completeness (*comp.*).

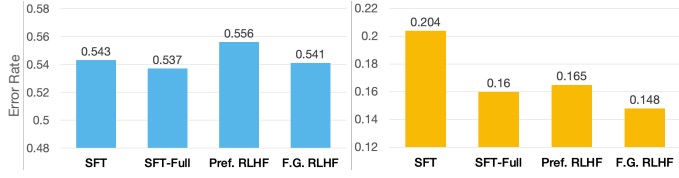

| Ours vs. | Win | Tie | Lose |
|---|---|---|---|
| SFT | **23.0%** | 65.5% | 11.5% |
| SFT-Full | **22.0%** | 61.0% | 17.0% |
| Pref. RLHF | **19.5%** | 71.0% | 9.5% |

Figure 3: Human evaluation on *rel.* (left) and *fact.* (right) error, measured by % of sub-sentences that contain the error type ($\downarrow$).

Table 2: Human pairwise comparison on information completeness (*comp.*) , where win/lose refers to FINE-GRAINED RLHF.

**FINE-GRAINED RLHF outperforms SFT and Preference RLHF on all error types.** Figure 3 and Table 2 show that our FINE-GRAINED RLHF leads to generation that is much more factually correct and contains more complete information, compared to all other systems. It generates fewer irrelevance, repetition, andincoherence errors, compared with SFT and Preference RLHF. In the meantime, Preference RLHF, despite greatly reducing factual errors compared to the initial policy model SFT, generates even more irrelevance, repetition, and incoherence errors than SFT. FINE-GRAINED RLHF outperforms Preference RLHF potentially due to more specific and localized training signals. In addition, we ask annotators to compare the overall generation quality of FINE-GRAINED RLHF and preference RLHF. Although Preference RLHF is trained directly with such preference feedback, FINE-GRAINED RLHF was rated better than Preference RLHF in 30.5% of all examples and worse in 24.5% of examples. The annotators indicate a tie in the remaining 45% of cases. Surprisingly, FINE-GRAINED RLHF outperforms SFT-Full with more factual and complete generation, despite a much lower annotation cost.

**RLHF is particularly effective in reducing factual errors.** Figure 3 shows that both FINE-GRAINED RLHF and Preference RLHF are effective in reducing factual errors in model generation. Meanwhile, we see little or no improvement in reducing irrelevance, repetition, or incoherence errors. We provide more in-depth analysis for this observation in §4.5.

Table 3 shows automatic scores on the QA-FEEDBACK test set, which show similar trends as human evaluation in terms of system comparisons, while all four systems achieve similar Rouge scores.

| | rel. $R_{\phi_1}(\uparrow)$ | fact. $R_{\phi_2}(\uparrow)$ | comp. $R_{\phi_3}(\uparrow)$ | Rouge($\uparrow$) |
|---|---|---|---|---|
| SFT-Full | 0.508 | 0.756 | 0.044 | 49.63 |
| SFT | **0.513** | 0.749 | -0.053 | 48.96 |
| + Pref. RLHF | 0.482 | 0.781 | 0.101 | 49.84 |
| + F.G. RLHF | **0.513** | 0.816 | 0.139 | **49.93** |

Table 3: Automatic evaluation on the QA-FEEDBACK test set.

| Config | rel. $R_{\phi_1}(\uparrow)$ | fact. $R_{\phi_2}(\uparrow)$ | comp. $R_{\phi_3}(\uparrow)$ | Rouge($\uparrow$) | avg. len |
|---|---|---|---|---|---|
| Short | **0.637** | 0.760 | -0.231 | 48.99 | 74.92 |
| Medium | 0.513 | 0.816 | 0.139 | **49.93** | 98.66 |
| Long | 0.425 | **0.860** | 0.241 | 48.72 | 109.63 |

Table 4: Automatic evaluation results (test set) of FINE-GRAINED RLHF trained with different reward model weight configurations.

## 4.5 LM Customization with FINE-GRAINED RLHF

Since we use multiple reward models in FINE-GRAINED RLHF, adjusting their weights (see Eq. 1) during RL may lead to different LM behaviors. For example, adding more weight to a reward model associated with one specific desired behavior type (e.g., information completeness) may lead the generation more towards that behavior type compared to others (e.g., information relevance). This flexibility can potentially fit users with diverse needs. Therefore, in this section, we explore FINE-GRAINED RLHF's ability to customize the LM behavior.

**LM customization.** As in Table 4, we explore three configurations of reward model weights ($w_1$, $w_2$, and $w_3$ for $R_{\phi_1}$, $R_{\phi_2}$, and $R_{\phi_3}$) and name them 'short', 'medium', and 'long' according to the LM's average generation length. For simplicity, we fix $w_2 = 0.5$ and $w_3 = 0.3$, and use 0.4, 0.3, and 0.2 for $w_1$, which leads to 'short', 'medium', and 'long' generation outputs respectively. We manually inspect 30 random examples and observe that (1) 'short' generates more relevant content, but is less factual and complete; (2) 'long', in contrast, gives the most factual and complete generation. This reflects that the LM is referencing a large amount of content from passages; (3) The 'medium' configuration balances the three rewards and has the highest Rouge score. 24/30 examples follow the above rule. Qualitative analysis and examples of LM customization are in Appendix A.

**Trade-off between error types.** We observe that a higher $w_1$ leads to a bigger *rel.* reward, smaller *fact.* and *comp.* rewards, and shorter generated outputs. One interpretation is that $R_{\phi_1}$ penalizes text spans that are irrelevant to the questions. As such, it encourages answering the question directly and penalizes referencing passages and generating auxiliary information. This reduces the model generation length and information completeness, and induces more factual errors.

## 4.6 Analysis

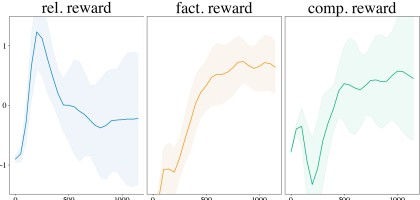

Figure 4: Dynamics of each type of reward during training (reward vs. training steps). All rewards are z-normalized.

| | rel. $R_{\phi_1}(\uparrow)$ | fact. $R_{\phi_2}(\uparrow)$ | comp. $R_{\phi_3}(\uparrow)$ | Rouge($\uparrow$) | avg. len |
|---|---|---|---|---|---|
| SFT | 0.514 | 0.735 | 0.065 | 43.13 | 96.69 |
| **F.G. RLHF** | 0.516 | **0.825** | 0.266 | **44.29** | 101.76 |
| w/o. $R_{\phi_1}$ | 0.249 | 0.771 | **0.742** | 38.52 | 179.31 |
| w/o. $R_{\phi_2}$ | **0.716** | 0.640 | -0.177 | 43.18 | 78.08 |
| w/o. $R_{\phi_3}$ | 0.565 | 0.799 | 0.123 | 43.61 | 93.92 |

Table 5: Ablation of reward models on the development set. $R_{\phi_1}$, $R_{\phi_2}$, and $R_{\phi_3}$ correspond to the reward model for relevance, factuality, and information completeness.

**Reward models are competing against each other.** In the prior section, we find that there is a trade-off between error types. To further look into this phenomenon, we explore the dynamics of each reward model during training. Figure 4 shows each reward model's rewards on the development set during training. All rewards are z-normalized for visualization. We see that the *fact.* reward is consistently increasing. The *rel.* reward increases rapidly in the first 250 steps and then starts decreasing, while the *comp.* reward exhibits an opposite trend, decreasing at first and then starting to increase. As discussed earlier, one interpretation is that relevance (precision) and information completeness (recall) can be adversarial objectives, so the rewards are competing. The three rewards reach an equilibrium point in later steps.

**Ablation: Does the LM learn from all reward models? What if we remove one reward model?** Table 5 explores the policy LM behavior when one of the three reward models is removed during training. Qualitative examples are in Appendix A. First, we observe that the corresponding reward decreases dramatically when the model is removed. When the *rel.* reward model ($R_{\phi_1}$) is removed,

the outputs become extremely long and the *comp.* reward is extremely high. We observe the outputs and find the model is copying a lot of content from the passages. When the *fact.* reward model ($R_{\phi_2}$) is removed, the *rel.* reward becomes the highest. We observe that the LM tends to answer the question directly and not reference the passages, which causes a lot of hallucinations. When the *comp.* reward model ($R_{\phi_3}$) is removed, the outputs are concise and factual but not providing all relevant information to the question. Thus, it has lower information completeness and Rouge score compared with the LM trained with all reward models.

**Reward model performance.** We report and analyze the performance of each reward model in predicting its corresponding error category. The *rel.* reward model $R_{\phi_1}$ has a binary classification accuracy of 69.6, and an F1 score (for the "`has error`" class) of 68.5 on model-generated sub-sentences from the development set. We sample 20 sub-sentences where $R_{\phi_1}$ predicts the opposite of the human label, and observe that all of them either 1) contain relevant auxiliary information and are marked as "`no error`" by humans, or 2) are marked as irrelevant by humans but provide closely related background information to the question. In other words, $R_{\phi_1}$ is mostly struggling with predicting the relevance of auxiliary information, and it rarely fails to predict a direct answer as "`no error`".

The *fact.* reward model $R_{\phi_2}$ has an accuracy of 77.8 and an F1 score of 67.5. We sample 20 sentences where $R_{\phi_2}$ makes a prediction mistake and we observe that the mistakes often happen when the generated sentence is highly abstractive instead of directly copying information from the passage. We also observe that more than 80% of human labeled factual errors occur when the model generates a direct answer (not auxiliary information) that contains hallucinated information or a random entity from a passage. We notice that $R_{\phi_2}$ correctly captures more than 80% of such errors.

The *comp.* reward model $R_{\phi_3}$ has an accuracy of 70.9 in pairwise comparison. In contrast, the preference-based reward model only reaches an accuracy of 57.2. This helps confirm our intuition that assessing long-form generation outputs holistically can be more ambiguous and subjective than evaluating the outputs with a focus on a specific undesired behavior type.

**Comparison with ChatGPT responses.** We experiment with answering the questions with ChatGPT. To familiarize ChatGPT with the style of our LFQA task, we prompt it with the task instruction and a single random QA example (due to length limitation). ChatGPT achieves a RougeLSum score of 40.92 on the test set, which is much lower than our models. We do not use our trained reward models to evaluate ChatGPT outputs because reward models trained on T5-large may not generalize well to ChatGPT. We instead manually inspect the ChatGPT responses, and observe that they are mostly concise and factual, yet lack the auxiliary information necessary to clarify ambiguous questions. Qualitative examples are in Appendix A. This shows the difficulty for ChatGPT in learning user-desired behaviors through simple prompting.

## 5 Related Work

**Reinforcement learning from human feedback (RLHF).** RLHF [46, 42, 29] aims to optimize the policy language model to generate content that is desired by human. This framework has been explored to improve the model performance on a variety of natural language processing tasks such as text summarization [40], instruction following [29], question answering [24, 27] and reducing harmfulness [3, 2, 22, 10]. Most of these studies collect human preferences over pairs of model outputs on one or a set of desired attributes, in order to train a reward model to assign a holistic score for a generation output during RL training. [13] trains separate reward models that assign scores for different desired attributes, but still uses a single reward that combines scores from all reward models. In contrast, we explore RLHF with fine-grained reward models trained on human feedback where each reward model provides dense reward after every small text segment for a specific type of desired behavior. [30] explores using intermediate rewards to improves LM performance on reasoning tasks.

**Learning from human feedback in NLP.** There also exists work that explores non-RL methods to learn from human feedback. [44] trains a reward model that predicts a single score for each model output and selects samples with the highest reward scores for supervised fine-tuning. [38, 14, 42] train a conversational model to predict both the response and a binary user satisfaction score in order to improve the response generation. Besides such numerical human feedback, natural language (NL) human feedback has also been explored. [23, 6] collect and store NL human feedback in a feedback memory for the model to retrieve and then perform the end task conditioning on the retrieved feedback. [5, 35, 34] use a refinement model to refine model outputs conditioning on NL human feedback and

then use a reward model to select the best refined outputs for supervised fine-tuning. Methods for using a reward model to guide LM generation towards desired behaviors at inference time [21, 7] can complement our work that aims to improve the LM during training. [16] also explores incorporating human feedback into LM pre-training.

# 6 Discussion

**Annotation Costs.** It is important to note that the fine-grained human feedback used for training our fine-grained reward models does *not* incur a greater cost than holistic human preference. As outlined in § 4.2, our observations reveal that annotators require a substantial amount of time to compare two lengthy text outputs. For the long-form QA task, both fine-grained feedback and preference-based feedback takes approximately 6 minutes per sample for an annotator.

## 6.1 Broader Impacts

We propose the FINE-GRAINED RLHF framework that can incorporate multiple reward models to provide dense rewards for RL training, which leads to LM outputs that are optimized towards such rewards. Our framework can be applied to any text generation task, thereby enhancing LM performance by offering more nuanced guidance than holistic feedback. The key advantages of the FINE-GRAINED RLHF framework are two-fold:

**Flexibility.** Our framework significantly expands the versatility of reward models for RLHF. For example, future work involving fact-checking, sentiment classification, toxicity detection, among others, can all be incorporated within this framework. LMs can be trained against all these reward models via FINE-GRAINED RLHF.

**Controllablility.** Having multiple reward models that stand for different feedback types allows the end user to exert greater control over RL training (e.g., through different combinations of reward model weights; see details in § 4.5). This leads to customized model behaviors, a benefit particularly valuable for applications like educational tools where model personalization is crucial.

## 6.2 Limitations and Future Work

One major limitation of our framework comes from the additional compute cost of getting *fine-grained* rewards, compared to RLHF with a holistic reward. For instance, in the detoxification task, we need to make multiple PERSPECTIVE API calls for each model output depending on how many sentences are generated, while RLHF with a holistic reward only requires one. In the long-form QA task, we need to calculate a dense reward from multiple reward models, which takes more compute time and GPU memory than a single reward model.

Another limitation is that different tasks may have different definitions of fine-grained feedback in terms of the feedback types and the density level of each type. Therefore, defining feedback that is well-suited for a task and training reward models accordingly requires non-trivial manual effort.

Finally, in this work, we carefully control the quality of annotated feedback, which is then used to train reward models for RL. In practice, when a deployed model is released to the public, end users don't always give clean feedback. Therefore, how to obtain effective learning signals from noisy human feedback in the wild still needs further investigation.

Some other interesting questions to explore in the future include: 1) Can we obtain fine-grained feedback from LMs like GPT-4 instead of humans to improve model performance and reduce annotation costs? 2) How can other non-RL approaches of using human feedback such as controlled generation during inference time complement FINE-GRAINED RLHF? 3) How would fine-grained reward and value model sizes affect policy model performance during RL training?

# 7 Conclusion

In this work, we introduce FINE-GRAINED RLHF, a framework that enables LMs to learn from multiple fine-grained reward models trained from human feedback, where each reward model detects a specific error category and provides dense rewards. We conduct experimental analysis on two text generation tasks to illustrate the performance gain of FINE-GRAINED RLHF than RLHF over holistic rewards, supported by both automatic and human evaluation. Furthermore, we show that an LM can be customized for specific needs using different combinations of fine-grained reward models.

## Acknowledgments

We thank Jiacheng Liu for sharing the standard PPO training code, and Yizhong Wang for providing insights during early discussions of the project. We also thank UW TIAL members for participating in our pilot feedback annotation. We extend our thanks to UW NLP members who provided insights or feedback to our project. Lastly, we especially thank all our AMT workers for helping us annotate the high quality feedback data. This research was developed with funding from the Defense Advanced Research Projects Agency (DARPA) under Contract No. FA8650-23-C-7316. This work was also funded in part by the DARPA MCS program through NIWC Pacific (N66001-19-2-4031), NSF IIS-2044660, and ONR N00014-18-1-2826. The views, opinions and/or findings expressed are those of the author and should not be interpreted as representing the official views or policies of the Department of Defense or the U.S. Government.

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

# Appendices

# A    Qualitative Examples for Long-Form QA

## A.1    Examples on LM Customization

As discussed in § 4.5, we can modify the weight of each fine-grained reward model during RL training to get LM with different behaviors. Here, we explore three configurations of reward model weights and name them 'short', 'medium', and 'long' based on the LM's average generation length. The 'short' configuration generates concise and short responses, while the 'long' configuration generates detailed and long responses. Table 6 demonstrates the different behaviors of our customized LMs. Given the same question, each LM generates different amount of auxiliary information in the response.

| Question: | When did the French join revolution on colonists' side? |
|---|---|
| Passages: | **Article Title: France in the American Revolutionary War**
French involvement in the American Revolutionary War began in 1775, when France, a rival of the British Empire, secretly shipped supplies to the Continental Army. A Treaty of Alliance in 1778 soon followed, which led to shipments of money and material to the United States. Subsequently, the Spanish Empire and the Dutch Republic also began to send assistance, leaving the British Empire with no allies. France's help is considered a vital and decisive contribution to the United States' victory against the British. As a cost of participation in the war, France accumulated over 1 billion livres in debt. After its defeat in the Seven Years' War in 1763, France lost its vast holdings in North America. Meanwhile, the American colonists and the British government began to fight over whether Parliament in London or the colonial assemblies had primary responsibility for taxation. As part of that conflict, the colonists organized the Boston Tea Party in response to a tax on tea. The British government responded by passing the Intolerable Acts, which included the closing of Boston Harbor and the revocation of Massachusetts's colonial charter. This conflict exacerbated tensions further. The ideological conflict escalated into open warfare in 1775, at which point the American patriots revolted against British rule. France, who had been rebuilding their Navy and other forces, saw this as an opportunity to seriously weaken her perennial enemy.
**Article Title: France in the American Revolutionary War**
France bitterly resented its loss in the Seven Years' War and sought revenge. It also wanted to strategically weaken Britain. Following the Declaration of Independence, the American Revolution was well received by both the general population and the aristocracy in France. The Revolution was perceived as the incarnation of the Enlightenment Spirit against the "English tyranny." Benjamin Franklin traveled to France in December 1776 in order to rally the nation's support, and he was welcomed with great enthusiasm. At first, French support was covert: French agents sent the Patriots military aid (predominantly gunpowder) through a company called "Rodrigue Hortalez et Compagnie", beginning in the spring of 1776. Estimates place the percentage of French supplied arms to the Americans in the Saratoga campaign at up to 90%. By 1777, over five million "livres" of aid had been sent to the American rebels. Motivated by the prospect of glory in battle or animated by the sincere ideals of liberty and republicanism, volunteers like Pierre Charles L'Enfant joined the American army. The most famous was Lafayette, a charming young aristocrat who defied the king's order and enlisted in 1777 at age 20. He became an aide to George Washington and a combat general. More importantly, he solidified a favorable American view of France.
**Article Title: Treaty of Alliance (1778)**
On March 17, 1778, four days after a French ambassador informed the British government that France had officially recognized the United States as an independent nation with the signing of the Treaty of Amity and Commerce and the Treaty of Alliance, Great Britain declared war on France, thereby engaging that nation in the American Revolutionary War. French entry into the war would lead to further escalation of the war when Spain entered the fight against Britain as France's ally, after the signing of the Treaty of Aranjuez on April 12, 1779, and again in December 1780 when Britain declared war on the Dutch Republic after seizing a Dutch merchant ship they claimed was carrying contraband to France during the Affair of Fielding and Bylandt. After the signing of the treaty French supplies of arms, ammunition, and uniforms proved vital for the Continental Army, while their actions in the Anglo French War of 1778-83 in the West Indies and elsewhere forced Britain to redeploy troops and naval units away from the North American colonies to secure their holdings. |

**LM outputs with different reward model configurations:**

| Short: | French involvement in the American Revolutionary War began in 1775, when France, a rival of the British Empire, secretly shipped supplies to the Continental Army. A Treaty of Alliance in 1778 soon followed, which led to shipments of money and matériel to the United States. Subsequently, the Spanish Empire and the Dutch Republic also began to send assistance, leaving the British Empire with no allies. |
|---|---|
| Medium: | French involvement in the American Revolutionary War began in 1775, when France, a rival of the British Empire, secretly shipped supplies to the Continental Army. A Treaty of Alliance in 1778 soon followed, which led to shipments of money and matériel to the United States. Subsequently, the Spanish Empire and the Dutch Republic also began to send assistance, leaving the British Empire with no allies. On March 17, 1778, four days after a French ambassador informed the British government that France had officially recognized the United States as an independent nation, Great Britain declared war on France, thereby engaging that nation in the American Revolutionary War. |
| Long: | French involvement in the American Revolutionary War began in 1775, when France, a rival of the British Empire, secretly shipped supplies to the Continental Army. A Treaty of Alliance in 1778 soon followed, which led to shipments of money and matériel to the United States. Subsequently, the Spanish Empire and the Dutch Republic also began to send assistance, leaving the British Empire with no allies. On March 17, 1778, four days after a French ambassador informed the British government that France had officially recognized the United States as an independent nation with the signing of the Treaty of Amity and Commerce and the Treaty of Alliance, Great Britain declared war on France, thereby engaging that nation in the American Revolutionary War. |

Table 6: Qualitative examples on LM customization.

## A.2 Examples on LM Errors

Table 7 and Table 8 show examples of LM outputs from all the compared systems (SFT, Pref. RLHF, and FINE-GRAINED RLHF). We mark the fine-grained errors on the model outputs. Overall, our FINE-GRAINED RLHF outperforms SFT and Pref. RLHF in all three error types.

| Question: | Who was the prime minister of Canada in 1920? |
|---|---|
| **Passages:** | **Article Title: Robert Borden** |
| | Not only did Borden's persistence allow him to represent Canada in Paris as a nation, it also ensured that each of the dominions could sign the Treaty of Versailles in its own right, and receive a separate membership in the League of Nations. During the conference Borden tried to act as an intermediary between the United States and other members of the British Empire delegation, particularly Australia and New Zealand over the issue of Mandates. Borden also discussed with Lloyd George, the possibility of Canada taking over the administration of Belize and the West Indies, but no agreement was reached. At Borden's insistence, the treaty was ratified by the Canadian Parliament. Borden was the last Prime Minister to be knighted after the House of Commons indicated its desire for the discontinuation of the granting of any future titles to Canadians in 1919 with the adoption of the Nickle Resolution. In 1919 Borden approved the use of troops to put down the Winnipeg general strike, which was feared to be the result of Bolshevik agitation from the Soviet Union. Sir Robert Borden retired from office in 1920. He was the Chancellor of Queen's University from 1924 to 1930 and also was Chancellor of McGill University from 1918 to 1920 while still Prime Minister. Borden also served as Vice-President of The Champlain Society between 1923 and 1925. He was the Society's first Honorary President between 1925 and 1938. |
| | **Article Title: Robert Borden** |
| | Sir Robert Laird Borden, (June 26, 1854 – June 10, 1937) was a Canadian lawyer and politician who served as the eighth Prime Minister of Canada, in office from 1911 to 1920. He is best known for his leadership of Canada during World War I. Borden was born in Grand-Pré, Nova Scotia. He worked as a schoolteacher for a period and then served his articles of clerkship at a Halifax law firm. He was called to the bar in 1878, and soon became one of Nova Scotia's most prominent barristers. Borden was elected to the House of Commons of Canada in 1896, representing the Conservative Party. He replaced Charles Tupper as party leader in 1901, and became prime minister after the party's victory at the 1911 federal election. As prime minister, Borden led Canada through World War I and its immediate aftermath. His government passed the "War Measures Act", created the Canadian Expeditionary Force, and eventually introduced compulsory military service, which sparked the 1917 conscription crisis. On the home front, it dealt with the consequences of the Halifax Explosion, introduced women's suffrage for federal elections, and used the North-West Mounted Police to break up the 1919 Winnipeg general strike. |
| | **Article Title: Arthur Meighen** |
| | Arthur Meighen Arthur Meighen (; 16 June 1874 – 5 August 1960) was a Canadian lawyer and politician who served as the ninth Prime Minister of Canada, in office from July 1920 to December 1921 and again from June to September 1926. He led the Conservative Party from 1920 to 1926 and from 1941 to 1942. Meighen was born in rural Perth County, Ontario. He studied mathematics at the University of Toronto, and then went on to Osgoode Hall Law School. After qualifying to practice law, he moved to Portage la Prairie, Manitoba. Meighen entered the House of Commons of Canada in 1908, aged 34, and in 1913 was appointed to the Cabinet of Prime Minister Sir Robert Borden. Under Borden, Meighen served as Solicitor General (1913–1917), Secretary of State for Canada (1917), Minister of Mines (1917; 1919–1920), Minister of the Interior (1917–1920), and Superintendent-General of Indian Affairs (1917–1920). In 1920, Meighen succeeded Borden as Conservative leader and Prime Minister – the first born after Confederation, and the only one from a Manitoba riding. He suffered a heavy defeat in the 1921 election to Mackenzie King and the Liberal Party, but re-entered Parliament through a 1922 by-election and remained as Opposition Leader. |

**LM outputs**

| | |
|---|---|
| **SFT:** | Sir Robert Borden was the eighth Prime Minister of Canada from 1911 to 1920. Arthur Meighen was the ninth Prime Minister of Canada from 1920 to 1926. |
| **Pref. RLHF:** | Sir Robert Laird Borden was a Canadian lawyer and politician who served as the eighth Prime Minister of Canada, in office from 1911 to 1920. Arthur Meighen was a Canadian lawyer and politician who served as the ninth Prime Minister of Canada, in office from July 1920 to December 1921 and again from June to September 1926. He led the Conservative Party from 1920 to 1926 and from 1941 to 1942. |
| **FINE-GRAINED RLHF** | Sir Robert Laird Borden was a Canadian lawyer and politician who served as the eighth Prime Minister of Canada, in office from 1911 to 1920. Arthur Meighen was a Canadian lawyer and politician who served as the ninth Prime Minister of Canada, in office from July 1920 to December 1921 and again from June to September 1926. |

Table 7: Examples of errors in LM outputs of SFT, Pref. RLHF, and our FINE-GRAINED RLHF. We mark the $C_1$ (irrelevance, repetition, or incoherence) and $C_2$ (incorrect or unverifiable facts) types of errors in the model outputs.

| Question: | Where does the flat iron cut of meat come from? |
|---|---|
| **Passages:** | **Article Title: Flat iron steak** |
| | Flat iron steak (US), butlers' steak (UK), or oyster blade steak (Australia and New Zealand) is a cut of steak cut with the grain from the shoulder of the animal. This produces a flavorful cut that is a bit tough because it contains a gristly fascia membrane unless removed. Some restaurants offer it on their menu, often at lower price than the more popular rib-eye and strip steaks of the same grade. This is used, in some places, as a means of selling a less expensive cut from the same animal, for example Kobe beef. This cut of steak is from the shoulder of a beef animal. It is located adjacent to the heart of the shoulder clod, under the seven or paddle bone (shoulder blade or scapula). The steak encompasses the infraspinatus muscles of beef, and one may see this displayed in some butcher shops and meat markets as a "top blade" roast. Anatomically, the muscle forms the dorsal part of the rotator cuff of the steer. This cut is anatomically distinct from the shoulder tender, which lies directly below it and is the teres major. Flat iron steaks usually have a significant amount of marbling. |

**LM outputs**

| **SFT:** | Flat iron steak is a cut of steak cut with the grain from the shoulder of the animal. This produces a flavorful cut that is a bit tough because it contains a gristly fascia membrane unless removed. Some restaurants offer it on their menu, often at lower price than the more popular rib-eye and strip steaks of the same grade. This is used, in some places, as a means of selling a less expensive cut from the same animal, for example Kobe beef. |
|---|---|
| **Pref. RLHF:** | Flat iron steak is a cut of steak cut with the grain from the shoulder of the animal. This produces a flavorful cut that is a bit tough because it contains a gristly fascia membrane unless removed. Some restaurants offer it on their menu, often at lower price than the more popular rib-eye and strip steaks of the same grade. This is used, in some places, as a means of selling a less expensive cut from the same animal, for example Kobe beef. |
| **FINE-GRAINED RLHF** | Flat iron steak is a cut of steak cut with the grain from the shoulder of the animal. This is located adjacent to the heart of the shoulder clod, under the seven or paddle bone (shoulder blade or scapula). This steak encompasses the infraspinatus muscles of beef, and one may see this displayed in some butcher shops and meat markets as a "top blade roast." |

Table 8: Qualitative examples on another question. Here we highlight the information in blue that is covered by the LM trained with FINE-GRAINED RLHF but missed ( $C_3$: incomplete information error) by SFT and Pref.RLHF.

## A.3 Examples on Reward Model Ablation

As discussed in § 4.6, reward models are competing against each other, and we experiment with removing one of the three reward models during RL training. Table 9 shows an example of how LMs behave in such scenarios. See § 4.6 for our observations.

| Question: | How much does it cost to raise the Costa Concordia? |
|---|---|
| Passages: | **Article Title: Costa Pacifica**
Costa Pacifica is a "Concordia"-class cruise ship for Costa Crociere. She was handed over to Costa Crociere on 29 May 2009. Her sister ships, "Costa Concordia" and "Costa Serena", were launched in 2006 and in 2007, with "Costa Favolosa" and "Costa Fascinosa" launched in 2011 and 2012 respectively. Onboard facilities include the Samsara Spa and PlayStation World, an area boasting PS3 consoles, and the first recording studio at sea. "Costa Pacifica" is the third ship of the "Concordia"-class, preceded by sister ships "Costa Concordia" and "Costa Serena" and was followed by "Costa Favolosa" and "Costa Fascinosa" on 2011 and 2012, all part of the expansion program of Costa which entailed an investment of 2.4 billion Euro, and is currently the largest fleet expansion program in the world. "Costa Pacifica" was ordered on 14 December 2005, by Carnival Corporation, the parent company of Costa Crociere. The order was four ships, which includes; "Carnival Dream", for Carnival Cruise Lines; "Nieuw Amsterdam" for Holland America Line; and "Ruby Princess" for Princess Cruises, at a total cost of US $3 billion.
**Article Title: Costa Concordia disaster**
On 13 January 2012, the Italian cruise ship "Costa Concordia" ran aground and overturned after striking an underwater rock off Isola del Giglio, Tuscany, resulting in 32 deaths. The eight year old Costa Cruises vessel was on the first leg of a cruise around the Mediterranean Sea when she deviated from her planned route at the Isola del Giglio, sailed closer to the island, and struck a rock formation on the sea floor. A six-hour rescue effort resulted in most of the passengers being brought ashore. "Costa Concordia" was officially declared a "constructive total loss" by the insurance company, and her salvage was "one of the biggest maritime salvage operations". On 16 September 2013, the parbuckle salvage of the ship began, and by the early hours of 17 September 2013, the ship was set upright on its underwater cradle. In July 2014, the ship was refloated by large sponsons (metal tanks) welded to its sides and was towed 320 kilometres (200 miles) to its home port of Genoa for scrapping which was finished in July 2017. The total cost of the disaster, including victims' compensation, refloating, towing and scrapping costs, is estimated at approximately $2 billion, more than three times the $612 million construction cost of the ship. |

**LM outputs: removing one fine-grained reward model during training**

| All reward models: | Costa Concordia is a cruise ship for Costa Crociere. On 13 January 2012, the Italian cruise ship Costa Concordia ran aground and overturned after striking an underwater rock off Isola del Giglio, Tuscany, resulting in 32 deaths. The total cost of the disaster, including victims' compensation, refloating, towing and scrapping costs, is estimated at approximately $2 billion, more than three times the $612 million construction cost of the ship. |
|---|---|
| **Without relevance RM** $R_{\phi_1}$: | On 13 January 2012, the Italian cruise ship Costa Concordia ran aground and overturned after striking an underwater rock off Isola del Giglio, Tuscany, resulting in 32 deaths. On 16 September 2013, the parbuckle salvage of the ship began, and by the early hours of 17 September 2013, the ship was set upright on its underwater cradle. In July 2014, the ship was refloated by large sponsons (metal tanks welded to its sides and was towed 320 kilometres (200 miles) to its home port of Genoa for scrapping which was finished in July 2017. The total cost of the disaster, including victims' compensation, refloating, refloating, towing and scrapping costs, is estimated at approximately $2 billion, more than three times the $612 million construction cost of the ship. The total cost of the disaster, including victims' compensation, refloating, towing and scrapping costs, is estimated at approximately $2 billion.
Comment: Too much irrelevant information & repetition error. |
| **Without factuality RM** $R_{\phi_2}$: | The cost to raise the Costa Concordia was estimated at approximately $2 billion, more than three times the $612 million construction cost of the ship.
Comment: Factuality error. Direct answer without any supporting evidence from the passages. |
| **Without info. completeness RM** $R_{\phi_3}$: | Costa Concordia is a "Concordia"-class cruise ship for Costa Crociere. She was handed over to Costa Crociere on 29 May 2009. The total cost of the disaster, including victims' compensation, refloating, towing and scrapping costs, is estimated at approximately $2 billion, more than three times the $612 million construction cost of the ship.
Comment: Miss the auxiliary information about the disaster. |

Table 9: Qualitative examples on the reward model ablation experiment. We show the LM outputs when we use all the reward models and when one of the reward models is removed during training. We also mark the errors in each output.

## A.4 Comparison with ChatGPT responses

We compare the responses generated by ChatGPT (one-shot)[3] and our system in Table 10. As discussed in § 4.6, We find that ChatGPT responses are relevant and factual, yet lack the auxiliary information to answer the ambiguous questions. This shows that it is challenging for ChatGPT to learn user-desired behaviors through prompting and in-context learning.

---

[3]Since the input for each example is very long, we cannot fit more than one in-context example into the model.

| Question: | When did the Rams go to St Louis? |
|---|---|
| Passages: | **Article Title: History of the Los Angeles Rams**
The Los Angeles Rams are a professional American football team that play in the National Football League (NFL). The Rams franchise was founded in 1936 as the Cleveland Rams in the short-lived second American Football League before joining the NFL the next year. In 1946, the franchise moved to Los Angeles. The Rams franchise remained in the metro area until 1994, when they moved to St. Louis, and were known as the St. Louis Rams from 1995 to 2015. The Rams franchise returned to Los Angeles in 2016. This article chronicles the franchise's history during their time in Los Angeles, from playing at the Los Angeles Memorial Coliseum between 1946 and 1979, to playing at Anaheim Stadium (now known as Angel Stadium of Anaheim) in Anaheim from 1980 to 1994, and its return to Southern California beginning with the season. On January 12, 1946, Dan Reeves was denied a request by the other National Football League (NFL) owners to move his team, the Cleveland Rams to Los Angeles and the then-103,000-seat Los Angeles Memorial Coliseum. Reeves threatened to end his relationship with the NFL and get out of the professional football business altogether unless the Rams transfer to Los Angeles was permitted. A settlement was reached and, as a result, Reeves was allowed to move his team to Los Angeles.
**Article Title: History of the St. Louis Rams**
The professional American football franchise now known as the Los Angeles Rams played in St. Louis, Missouri, as the St. Louis Rams from the 1995 through the 2015 seasons before relocating back to Los Angeles where the team had played from the 1946 season to the 1994 season. The Rams franchise relocated from Los Angeles to St. Louis in 1995, which had been without a National Football League (NFL) team since the Cardinals moved to Phoenix, Arizona in 1988. The Rams' first home game in St. Louis was at Busch Memorial Stadium against the New Orleans Saints on September 10, 1995, before the Trans World Dome (later the Edward Jones Dome, and now known as The Dome at America's Center) was completed for their November 12 game against the Carolina Panthers. Their last game played at the Edward Jones Dome in St. Louis was against the Tampa Bay Buccaneers on December 17, 2015, which they won, 31–23. The Rams' last game as a St. Louis-based club was on January 3, 2016, against the San Francisco 49ers at Levi's Stadium, where they lost in overtime 19–16. |

**Comparison between our output and ChatGPT output**

| ChatGPT: | The Rams moved to St. Louis in 1995 and were known as the St. Louis Rams from 1995 to 2015. |
|---|---|
| FINE-GRAINED RLHF | The Los Angeles Rams are a professional American football team that play in the National Football League (NFL). The Rams franchise was founded in 1936 as the Cleveland Rams in the short-lived second American Football League before joining the NFL the next year. In 1946, the franchise moved to Los Angeles. The Rams franchise remained in the metro area until 1994, when they moved to St. Louis, and were known as the St. Louis Rams from 1995 to 2015. The Rams franchise returned to Los Angeles in 2016. |

| Question: | Who sang the song I'm Coming Out? |
|---|---|
| Passages: | **Article Title: I'm Coming Out**
There's also been a cover by Marcia Hines. It was featured as a cover for the video game "Karaoke Revolution" as well. In more recent times, the song was sampled in Keyshia Cole's remake of "I'm Coming Out" (featuring Iggy Azalea) for the original soundtrack of the 2014 movie The Other Woman, starring Cameron Diaz, Kate Upton and Leslie Mann. The song is sampled in "Break Your Heart Right Back" from Ariana Grande's sophomore album "My Everything" (2014). This song was also covered for the soundtrack to the 2016 movie "Trolls". In 1997, German comedian Hella von Sinnen sampled "I'm Coming Out" for her single "Mein Coming Out", using the term "coming out" in the context of coming out of the closet, or revealing one's homosexuality. In 2003, American singer Amerie covered "I'm Coming Out" for the soundtrack to the film "Maid in Manhattan". Her version was released as a single in select European countries and Australia. The final single version to be released was the Loren Dawson remix. The original single version (which is similar to the original) can be found on both CD singles and the original motion picture soundtrack to "Maid in Manhattan". Ross' version appears on the film's soundtrack album as well.
**Article Title: I'm Coming Out**
"I'm Coming Out" is a song recorded by American singer Diana Ross. It was written and produced by Chic members Bernard Edwards and Nile Rodgers, and released in August 22, 1980 as the second single from Ross' self-titled tenth album "Diana" (1980). In 1979, Ross commissioned Chic founders Nile Rodgers and Bernard Edwards to create material for a new album after taking her daughters to see the band in concert, following the success of Ross's final Ashford & Simpson-composed and produced LP," The Boss. "Rodgers got the idea for "I'm Coming Out" after noticing three different drag queens dressed as Diana Ross at a New York club called the GG Barnum Room. The lyrics hold an additional meaning to Ross, as she was leaving Motown Records and "coming out" from under Berry Gordy's thumb. According to the BBC4 documentary "How to Make It in the Music Business", Ross loved the record until she put it in front of Frankie Crocker, who pointed out that 'I'm Coming Out' is what homosexuals use to announce their homosexuality; she ran back to the studio in tears and demanded to know why Rodgers wanted to ruin her career. The song was another hit from the album, peaking at number five on the U.S. |

**Comparison between our output and ChatGPT output**

| ChatGPT: | The song "I'm Coming Out" was recorded by American singer Diana Ross. It was written and produced by Chic members Bernard Edwards and Nile Rodgers, and released in August 22, 1980 as the second single from Ross' self-titled tenth album "Diana" (1980). |
|---|---|
| FINE-GRAINED RLHF | "I'm Coming Out" is a song recorded by American singer Diana Ross. It was written and produced by Chic members Bernard Edwards and Nile Rodgers, and released in August 22, 1980 as the second single from Ross' self-titled tenth album Diana. In 2003, American singer Amerie covered "I'm Coming Out" for the soundtrack to the film Maid in Manhattan. Her version was released as a single in select European countries and Australia. The final single version to be released was the Loren Dawson remix. |

Table 10: Comparison between ChatGPT outputs and FINE-GRAINED RLHF outputs.

# B  Algorithm and Training Details of FINE-GRAINED RLHF

## B.1  PPO with Fine-Grained Rewards

The algorithm below shows in detail how PPO updates the policy LM $P_\theta$ and the value model $V_\psi$ with $K$ fine-grained reward models $R_{\phi_k}$.

---

**Algorithm 1** Fine-Grained Reinforcement Learning from Human Feedback (FINE-GRAINED RLHF)

---

**Input** initial policy model $P_{\theta_{\text{init}}}$; initial value model $V_{\psi_{\text{init}}}$; $K$ reward models $R_{\phi_k}$ trained from human feedback; task prompts $\mathcal{D}$; hyperparameters $\gamma$, $\lambda$, $\epsilon$, $\beta$      ▷ § 2

1:   policy model $P_\theta \leftarrow P_{\theta_{\text{init}}}$, value model $V_\psi \leftarrow V_{\psi_{\text{init}}}$
2: **for** step = 1, …, M **do**
3:      Sample a batch $\mathcal{D}_b$ from $\mathcal{D}$
4:      Sample output sequence $y^n \sim P_\theta(\cdot \mid x^n)$ for each prompt $x^n \in \mathcal{D}_b$
5:      Compute rewards $\{r_t^n\}_{t=1}^{|y^n|}$ for each sampled output $y^n$ by running $R_{\phi_k}$      ▷ Eq. 1
6:      Compute advantages $\{A_t\}_{t=1}^{|y^n|}$ and value targets $\{V^{\text{targ}}(s_t)\}_{t=1}^{|y^n|}$ for each $y^n$ with $V_\psi$
7:      **for** PPO iteration = 1, …, $\mu$ **do**
8:          Update the policy model by maximizing the PPO clipped surrogate objective:

$$\theta \leftarrow \arg\max_\theta \frac{1}{|\mathcal{D}_b|} \sum_{n=1}^{|\mathcal{D}_b|} \frac{1}{|y^n|} \sum_{t=1}^{|y^n|} \min\left( \frac{P_\theta(a_t \mid s_t)}{P_{\theta_{\text{old}}}(a_t \mid s_t)} A_t,\, \text{clip}(v_t,\, 1 - \varepsilon,\, 1 + \varepsilon) A_t \right)$$

9:          Update the value model by minimizing a square-error objective:

$$\psi \leftarrow \arg\min_\psi \frac{1}{|\mathcal{D}_b|} \sum_{n=1}^{|\mathcal{D}_b|} \frac{1}{|y^n|} \sum_{t=1}^{|y^n|} \left( V_\psi(s_t) - V^{\text{targ}}(s_t) \right)^2$$

**Output** $P_\theta$

---

## B.2  Implementation Details

**Model architectures.** For the detoxification experiments, the policy model is initialized with GPT2-large [31], and the value model is initialized with GPT2-base. For the long-form QA experiments, the policy model is initialized with a supervised fine-tuned T5-large [32], and the value model is initialized with T5-base. This design follows InstructGPT [29], which uses a larger (175B) policy model, and smaller value and reward (6B) models.

**Training details on detoxification.** For both the holistic reward baseline and the sentence-level (fine-grained) reward, we do a hyper-parameter search with the same set of hyper-parameters. For training, we run 200K episodes. The batch size (number of episodes per card during training) is 64. We use Adam optimizer with a linear learning rate scheduler and 10 warmup steps. We perform a hyper-parameter grid-search for peak learning rate $\in \{5e-6, 1e-5, 2e-5\}$, KL coefficient $\beta \in \{0.1, 0.2, 0.3\}$, discounting factor $\lambda \in \{0.95, 0.97, 0.99\}$, and the frequency of exploration (number of sampled outputs) $\in \{2, 4, 8\}$. We find that the higher the KL coefficient, the lower the perplexity, and the higher toxicity. This is consistent with findings from previous RLHF studies ([29], [33]). For a fair comparison, we eventually choose a set of parameters that achieve a similar level of perplexity for both reward models. The optimal set of hyper-parameters for holistic reward is $\beta = 0.3, \lambda = 0.99$. For sentence-level reward $\beta = 0.1, \lambda = 0.95$. The learning rate is $1e-5$, and the exploration frequency is $4$ for both experiments. We choose the checkpoint with the lowest validation set toxicity for evaluation. Regarding computation time, we use $2\times$ 80G NVIDIA A100 GPU for training, and the run time is about 22 hours.

**Training details on long-form QA.** We conduct a similar hyper-parameter grid search as our detoxification experiments. For long-Form QA, the input length limit is 1024, and the output length limit is 200. Notice that this is much longer than detoxification, so we use a smaller batch size and fewer training episodes. We experiment with multiple combinations of reward model weights. Fixing $w_2 = 0.5$ (factuality reward weight), we perform a grid search on $w_1, w_3 \in [0.0, 0.5]$.

We eventually choose $w_1 = 0.3, w_2 = 0.5, w_3 = 0.3$, which reaches a balance between three reward models and allows all three rewards to increase during training. For training, the batch size (number of episodes per card during training) is 32. We use Adam optimizer with a linear learning rate scheduler and 100 warmup steps. We perform a hyper-parameter grid-search for peak learning rate $\in \{5e-6, 1e-5, 2e-5\}$, KL coefficient $\beta \in \{0.1, 0.2, 0.3\}$, discounting factor $\lambda \in \{0.95, 0.97, 0.99\}$, and the frequency of exploration $\in \{2, 4, 8\}$. The optimal set of hyper-parameters for Pref. RLHF is $\beta = 0.2, \lambda = 0.99$. For FINE-GRAINED RLHF, $\beta = 0.3, \lambda = 0.95$. The learning rate is $1e-5$, and the exploration frequency is 4 for both experiments. we run 80K episodes, which is approximately 5 epochs. We choose the checkpoint with the highest validation reward for evaluation. Regarding computation time, we use $2 \times$ 80G NVIDIA A100 GPU for training, and the run time is about 15 hours.

**A note on the error bars.** All results we report in the paper are from 3 independent runs. The scores reported are all averaged across all runs. The error bars are represented as the shades behind each training curve in our figures. It shows the standard error across three runs.

## C   Long-Form QA Data and Human Feedback Annotation

### C.1   Data Construction

ASQA [39] is a long-form QA dataset that focuses on answering ambiguous factoid questions in an *open-domain* setting that requires passage retrieval from a given Wikipedia passage corpus. We reformulate it into a *reading comprehension* setting: given the input $x$ that contains a question $q$ and a set of knowledge passages $P = \{p_1, ..., p_{|P|}\}$, generate a long-form response $y$. To construct $P$ for each input $x$, we use the oracle knowledge contexts provided by ASQA for each $x$, that are text snippets from the passage corpus. We use BM25[4] to map each knowledge context (text snippet) to the closest passage from the passage corpus and use the resulting passages as $P$. Our train and dev examples come from the original ASQA train set and our test examples are the original ASQA dev examples.

### C.2   Human Feedback Annotation

**Fine-grained feedback.** As discussed in § 4.1, we first use 1K randomly sampled training examples to train a T5-large based supervised model **SFT** as the initial policy model $P_{\theta_{init}}$. Then we collect feedback on sampled outputs from SFT for the remaining 2,853 training examples and the 500 development examples, using the Amazon Machanical Turk platform.[5]

Figure 5 shows the fine-grained human feedback annotation interface with an example from QA-FEEDBACK. In addition to the task input—question $q$ and oracle passages $P$, we also provide a human-written response from ASQA to the worker as reference. However, it is important to note that, in practice, the annotation of our fine-grained feedback should not require the human-written response. The only purpose for us to provide the gold response is to have our workers follow the same question interpretation and expected response of the workers who annotate for ASQA, such that our experimental comparison with supervised models (**SFT** and **SFT-Full**; details in § 4.3) is fair. However, we still instruct our workers to strictly use the given passages for checking factual errors. For each span error, we ask the worker to select one out of 5 categories shown in Figure 6 (left).[6] However, we collapse these 5 categories into two categories ($C_1$ and $C_2$ mentioned in § 4.1) based on whether the error detection depends on the passages or not. When workers mark passage sentences as containing missing information, we instruct them to categorize each sentence as missing "answer", "major auxiliary information" or "minor auxiliary information," as shown in Figure 6 (right). Our instruction to the worker is provided in Figure 8.

**Quality control.** Before feedback collection, we design a qualification task to select qualified workers for this feedback annotation task. The qualification task consists of 5 questions with their corresponding passages and model outputs for the workers to annotate. We manually review about

---

[4]`https://github.com/castorini/pyserini`

[5]`https://www.mturk.com/`

[6]We see very few "incoherence" errors (1%), so the majority of labeled errors are from the other four categories during annotation.

70 submissions of the qualification task and select 15 workers whose annotation is marked by us as of high quality. Throughout the actual feedback annotation process, we constantly monitor the annotated data and send feedback to workers.

**Preference-based feedback.** For comparison purposes, we follow [29] to collect pairwise human preferences from the same group of workers we select from the qualification task. We sample four model outputs for each prompt $x$, which gives 6 pairs of model outputs. Similarly, we provide the worker with the human-written response and ask the workers to indicate pairwise preferences (ties are allowed) based on all errors they can find each model output. Figure 7 shows the preference-based human feedback annotation interface with an example from QA-FEEDBACK.

**Pay structure.** We pay a base rate of \$1.5 per example for annotating fine-grained or preference feedback. If the example consists of $\geq 3$ passages to read, we assign an additional \$0.3 bonus to the example. On average, we pay roughly \$1.65 per example for both tasks, which gives an \$16.5 hourly pay for our workers.

### C.3 Analysis of Collected Fine-Grained Feedback

Overall, among all *error spans* we collect, 76% of them are $C_1$ errors and the remaining 24% are $C_2$ errors. However, it is important to note that we instruct workers to label $C_2$ errors only at places that don't have a $C_1$ error. 75% examples are labeled as being incomplete; i.e., containing missing information that can be found in the given passages ($C_3$). Among all marked passage sentences that contain missing information, 31%, 42% and 27% are missing answer, major auxiliary information and minor auxiliary information respectively.

To analyze human-human agreement, a subset of 300 examples receive annotations from two distinct workers. We observe that while the exact agreement in error span boundaries is low, workers achieve reasonably high agreement on whether a sub-sentence contains $C_1$ (reach an agreement for 83% of all sub-sentences) and whether a sentence contains $C_2$ (92%). [7] The agreement on whether a model output contains complete information or not ($C_3$) is 85%. Therefore, we decide to have the density for error type $C_1$, $C_2$, and $C_3$ as sub-sentence, sentence and full sequence.

## D  Long-Form QA Reward Model Training Details

We train reward models with the 2,835 training examples with feedback collected and select the best model for each error category based on the their performance on the development set. The batch size and training epochs are 24 and 50 for $R_{\phi_1}$ and $R_{\phi_2}$. Each training is run on a single 80G NVIDIA A100 GPU, taking 1 and 2 hours for training $R_{\phi_1}$ and $R_{\phi_2}$ respectively.[8] The batch size and training epochs are 12 (per GPU) and 30 for $R_{\phi_3}$ and the preference-based reward model. Each training is run on $2\times$ 80G NVIDIA A100 GPU and takes 2 hours. We use Adam optimizer with a linear learning rate scheduler for all reward model training. For each reward model, we search the learning rate over $\{5e^{-6}, 1e^{-5}, 5e^{-5}\}$, weight decay over $\{0.001, 0.01\}$, and warm-up step ratio over $\{0.1, 0.2\}$ based on the dev set performance. Specifically, we use the model for $R_{\phi_1}$ and $R_{\phi_2}$ that achieve the best binary classification accuracy. For $R_{\phi_3}$ and the preference-based reward model, we select the model that achieves the best pairwise comparison accuracy. We also provide more training details for each reward model below.

$R_{\phi_1}$ **for** $C_1$**: Irrelevance, repetition, or incoherence.** To train the reward model $R_{\phi_1}$ that detects error of irrelevance, repetition, or incoherence, we apply a token-level classification loss to each [sep] token before $y_j^1$, where its gold label $g_j$ is "has error" if there is a $f_i \in \mathcal{F}$ that has $a_{b_i,\ldots,e_i}$ overlapped with $y_j^1$ and $c_i = 1$, and "no error" otherwise. We observe that most of the spans marked as error type $C_1$ that are shorter than 5 words usually carry very little information or are annotated as a result of workers being very careful or strict. Therefore, we filter out such short spans before constructing training examples for $R_{\phi_1}$. Overall, we get 7379 and 8059 sub-sentences with the "has error" and "no error" label respectively.

---

[7] We use spaCy [15] to segment generated model outputs into sentences. We then split sentences into sub-sentences using a comma or semicolon.

[8] Note that training $R_{\phi_1}$ takes shorter time as its input does not contain passages.

$R_{\phi_2}$ **for $C_2$: Incorrect or unverifiable facts.** We train $R_{\phi_2}$ in a similar way as how we train $R_{\phi_1}$. Instead of predicting the error for each sub-sentence, $R_{\phi_2}$ is trained to predict at the sentence level (i.e., $y_j^2$ is the $j^{th}$ sentence in $y$). Since workers do not annotate $C_2$ error for spans that are already labeled as having $C_1$ error, in order to avoid false negatives in training $R_{\phi_2}$, we do not provide gold label nor calculate loss for sentences that only contain $C_1$ error from training. In other words, all sentences that contain a $C_2$ error has the gold label "`has error`" and sentences that contain no error has the gold label "`no error`". Overall, we get 1600 and 3411 sentences with the "`has error`" and "`no error`" label respectively.

$R_{\phi_3}$ **for $C_3$: Incomplete information.** Instead of casting this as a classification task, $R_{\phi_3}$ predicts a single scalar reward given $x$ and $y$ and is trained with a pairwise comparison loss [29]. This is motivated by early work [19] that shows the better reliability of pairwise comparison than error classification when assessing a full generation sequence. To construct training data for $R_{\phi_3}$, we bootstrap pairwise comparisons from the corrected model output $y'$ as follows. We first map each sub-sentence in $y'$ to a passage sentence in $P$ that has a sub-string with the highest token-level F1 score with the sub-sentence,[9] and denote all mapped sentences as $S$. We then sample four responses from SFT, for each we do the same sentence mapping to get a set of passages sentences $S'$. We calculate $score(y) = |S' \cap S|/|S|$ as the information completeness score for each model response $y$. We follow [29] to pair up sampled responses for $q$ and denote each sampled response pair as $(\bar{y}_p, \bar{y}_l)$, where $score(\bar{y}_p) > score(\bar{y}_l)$. We drop the pairs where $score(\bar{y}_p) = score(\bar{y}_l)$. Then we follow [29] to train $R_{\phi_3}$ with the loss function in Eq. 2. We have a total number of 6821 pair examples in training.

**Preference-based reward model.** The preference-based reward model is trained in a similar way as $R_{\phi_3}$, with $\bar{y}_p$ representing the human preferred response against $\bar{y}_l$ in the loss function Eq. 2. We drop the pairs where a tie is indicated. We have a total number of 14981 pair examples in training.

---

[9]We manually review 50 mapped passage sentences and find over 90% of them are correctly mapped, which indicates frequent extractive behaviors from $P_{\theta_{init}}$.

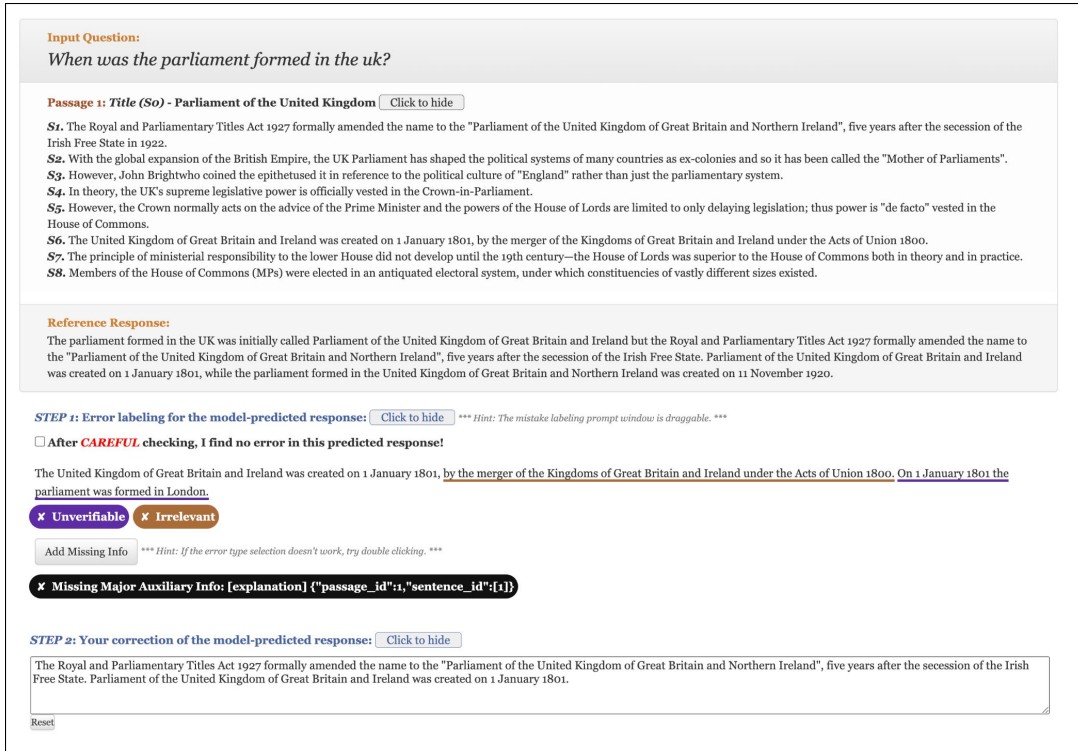

Figure 5: Fine-grained feedback annotation interface.

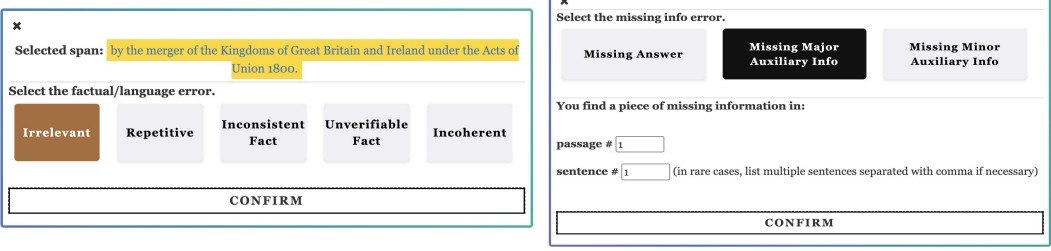

Figure 6: Error types in the fine-grained feedback annotation interface.

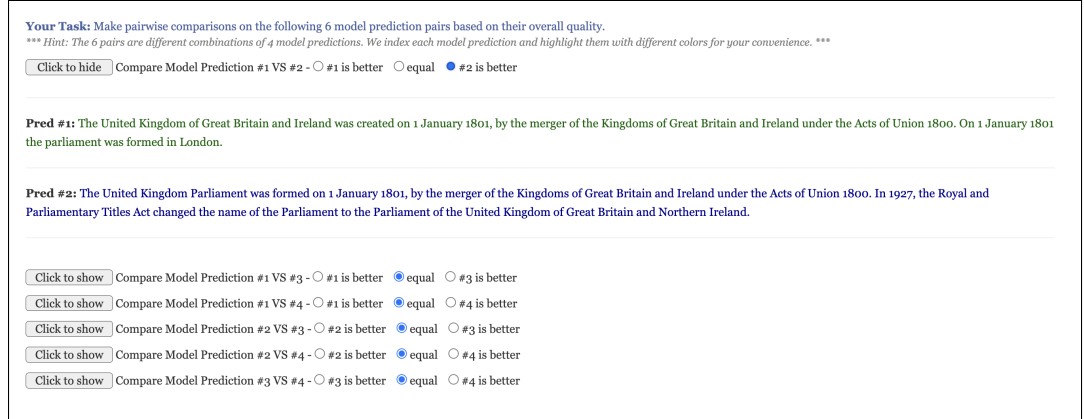

Figure 7: Preference feedback annotation interface. The task input is omitted (same as in Figure 5).

In each task, **you will be given** a question, a set of Wikipedia passages (with their article title provided), a reference response, and a model-predicted response. Each passage is presented as a sequence of sentences (title is indexed as sentence #0). **Your goal is to** 1) label mistakes made in the model prediction and 2) make corrective edits to the model prediction based on your labeled mistakes.

**Important Definitions**: *An ideal response* to a question should provide both ***answer(s)*** that directly responds to the question and ***crucial auxiliary information*** for better comprehension of the answer(s). We consider auxiliary information as *crucial* if it is used in the reference response. Additionally, all information in *an ideal response* should be ***factually consistent with (i.e., grounded in) the passages***. Note that the reference response is written by a human with potentially different grounding passages, and thus, you might find ***answers*** that can be found in the passages but are not included in the reference, which are ***STILL*** expected to be in the model prediction. On the other hand, answers in the reference that cannot be found in or verifiable by the passages are ***NOT*** expected to be in the model prediction. ***To conclude, all answers*** are expected in the model prediction ***IF AND ONLY IF*** it can be found in the passages. ***Crucial auxiliary information*** is expected in the model prediction ***IF AND ONLY IF*** it can be found in both the reference response and the passages.

Here are the detailed annotation steps:

***STEP 1 -*** Read the question and label mistakes made in a model-predicted response. As explained above, leverage the reference, **BUT** rely on the passages. Decide the mistake type and follow detailed instructions as follows. We encourage you to use CTRL+F/CMD+F for navigating the reference and passages using keywords.

- **Erroneous Span (i.e., substring)**: Highlight each span in the model prediction that contains one of the following errors. Label each span ***as short as possible*** and make sure each labeled span only contain ***one single*** information piece. You will be prompted to select the error type and provide an explanation if apply. For each span, label ***only one*** error type. If a span contains multiple errors, select the error type based on the order below (e.g., if a span contains "irrelevant" information, ignore any "inconsistent fact" it contains).

    i. **[Irrelevant]**: The span contains "irrelevant" information (e.g., neither an answer nor crucial auxiliary information, defined in the first 2 sentences in "*Important Definitions*"). To detect "irrelevant" errors, you do not need to consider whether the information is factually consistent with the passages or not.

    ii. **[Repetitive]**: The span repeats information in its previous text. Provide the previous text being repeated (***as short as possible***). ***Note that*** a "repetitive" span should still be marked even if its previous text being repeated contains an factual or coherence issue (defined below). However, if the previous text is "irrelevant", it should be marked as "irrelevant" too.

    iii. **[Inconsistent Fact]**: The span is factually inconsistent with the passages. Enter the passage id and sentence id(s) as evidence. ***Note that*** if you find multiple evidences in the passages, mark only one of them. The need for multiple passage ids usually indicates that you should separate the error into multiple ones (due to multiple information pieces).

    iv. **[Unverifiable Fact]**: The span is factually unverifiable (i.e., not mentioned in any passage), after ***carefully*** checking all passages. Common sense (e.g., "a bicyle has two wheels") doesn't need to be verified. However, do not count knowledge only commonly known in a specific region/community as commonsense. This can be subjective, and simply follow your best judgment.

    v. **[Incoherent]**: The span contains major grammar error (ignore minor typos), is uninterpretable, contradicts to common sense, or is not coherent with its context.

- **Missing Information**: Identify information that is expected but missing in the model prediction. Check "*Important Definitions*" above to see how to identify such information. Classify ***each piece*** of missing information as **[Missing Answer]** or **[Missing Major/Minor Auxiliary Information]**, and enter the passage id and sentence id(s) as evidence. Mark the missing auxiliary information as ***major*** if you think the information is indeed helpful for understanding the answer. Otherwise (e.g., a bit off-topic), mark it as ***minor***. Simply follow your best judgment. Follow the same ***"Note that"*** rule under *"[Inconsistent Fact]"* above.

***STEP 2 -*** Correct the model prediction to address your labeled mistakes in STEP 1. ***Instead of*** copying and pasting the reference into the box, make minimal local edits on the original prediction. Make sure the corrected version is fluent and free of typos. In rare cases when you find no useful information in passages for answering the question, you can leave the box as blank if necessary.

Figure 8: Fine-grained feedback annotation instructions.

