# OpenReview forum: "Fine-Grained Human Feedback Gives Better Rewards for Language Model Training"
_NeurIPS.cc/2023/Conference — NeurIPS 2023 spotlight_

### Official Review · Reviewer_uPU2 · 2023-07-06

**Soundness:** 3 good
**Presentation:** 3 good
**Contribution:** 3 good
**Rating:** 7
**Confidence:** 4

**Summary:**

This paper proposes *FINE-GRAINED RLHF*: an extension to (the now popular) *reinforcement learning from human feedback* (RLHF) that allows incorporating more granular human feedback into the language model (LM) training process. Typically, RLHF leverages reward/preference models that are trained to output a *scalar* reward for the whole generated sequence, and this *reward sparsity* can hinder improvements for long-sequence generation. Furthermore, scalar rewards do not offer insights into which parts of the sequence were problematic or displayed undesired behaviors.

To address this, the using *fine-grained reward models*: for each "category" of feedback of interest, a separate reward model is trained. Furthermore, these models generate feedback at lower granularity (for example, at sentence-level). The policy LM can then be optimized with respect *weighted* sum of the rewards at each timestep, using standard RL algorithms like PPO.

The authors explore the impact on two language generation tasks: (1) detoxification and (2) long-form question answering.
For detoxification, the authors find that (with a reward based on an external API) training with rewards provided for every sentence yields better performance when compared to a holistic sequence-level reward. However this evaluation is fully automatic.
For long-form question answering (QA), the authors introduce a novel dataset, QA-Feedback, accompanied with fine-grained human feedback on initial policy LM. The feedback comes in three error categories at different levels of granularity: (a) irrelevance, repetition or incoherence (sub-sentence level), (b) incorrect or unverifiable facts (sentence level), and (c) incomplete information (whole sequence level). They collect this feedback through crowd-workers. They also collect *preference-based* feedback, for comparison with (hollistic) RLHF.
By training separate reward models for each category, the authors show that fine-grained RLHF outperforms the traditional RLHF and supervised baselines, both in automatic and human evaluation. Interestingly, by carefully adjusting the weights of different reward models, the authors show LM behaviors can be fine-tuned to optimize different combinations of desired behavior, and outperform. The authors also explore the reward model accuracy and even compare against a ChatGPT baseline.

**Strengths:**

I really enjoyed reading this paper. It explores a very prominent topic recently (learning from human feedback), it is well written and it does an very extensive analysis (at least for one of the tasks it tackles). “Reward sparsity” is a well known problem in RL, and recent works in RLHF for LLMs just kinda ignores it. Their proposed fine-grained RLHF is seems like a good approach to tackle it, makes intuitive sense and seems easy to implement. Their findings surrounding controllability could also have significant implications for (RLHF’d) LLMs. They also introduce a novel dataset for QA with fine-grained feedback.

**Weaknesses:**

The main weakness of this paper was that only a single LLM was tried: while I believe that their findings will generalize to other LLMs, an experimental validation of this would make this paper even stronger. Even trying re-using the reward models trained for the initial GPT-2 policy on some other LLaMA would already be quite a bonus.

Also a couple of other minor ones:

- I think task 1 (detoxification) in general is not a good setup: in uses closed source APIs for model-based (from a relatively weak one) PPL as evaluation. the simulated “sentence-level” reward is also not very elegant. I would ideally suggest trying to replace it with another simple, controllable generation task and preferably using human-eval. But ye most of these are covered by task 2
- The multi-reward aspect of the algorithm makes RLHF now a sort of *multi-task* optimization problem, and bring with the problem of how to weight the different categories. The discussion on the trade-off in (4.5) is interesting and helpful, but I think more could be said on how to pick a weighting for a general case where we just want a single “quality”. Also a very simple pareto analysis (probably trying more weightings of the reward) and some plots showing the final trade-off (ideally in terms of human reward?) would add more technical insight

**Questions:**

- The levels of granularity in the paper make sense for the three categories considered, but have you thought about the more general problem of allowing arbitrary granularities? It seems it is mostly a UX/feedback collection process, and that nothing in the algorithm would prevent it.
- Is there plans to open-source the (fine-grained) RLHF code? Integrating this into popular packages for RLHF (like TRLX) could heavily boost the impact of this work

**Limitations:**

See above

---

> ### Author Rebuttal · Authors · 2023-08-09
>
> Thanks for your insightful comments! Please see our responses below.
>
> > ***1. The main weakness of this paper was that only a single LLM was tried: while I believe that their findings will generalize to other LLMs, an experimental validation of this would make this paper even stronger. Even trying re-using the reward models trained for the initial GPT-2 policy on some other LLaMA would already be quite a bonus.***
>
> We agree that analyzing the scaling of our framework to larger models would bring insights to the community, but **our work focuses on analyzing the effectiveness of using fine-grained reward for training LMs using RL**, compared to holistic reward. Due to limited compute resources (RL training is expensive), **we leave the exploration of model scaling for future work**. We do believe that, since many research groups and practitioners face similar computational budget issues, the findings we report on smaller models will be of wide interest.
>
>
>
> > ***2. Also a couple of other minor ones:***
>
> > I think task 1 (detoxification) in general is not a good setup: in uses closed source APIs for model-based (from a relatively weak one) PPL as evaluation. the simulated “sentence-level” reward is also not very elegant. I would ideally suggest trying to replace it with another simple, controllable generation task and preferably using human-eval. But ye most of these are covered by task 2
>
> Our setup of task 1 including the policy model (GPT2) and evaluation metrics are consistent with previous studies of our baselines. Although it’s not perfect, it leads to a fair comparison with the baselines. The use of a reward at the sentence-level provides a contrasting case of “fine-grained” compared to task 2. The specific definition for the sentence-level reward (the score change with a newly generated sentence) was carefully designed based on the definition of PERSPECTIVE scores (the likelihood that someone will perceive the text sequence as toxic) and reward in RL (the reward collected by taking the action $a$ at state $s$).
>
>
> > The multi-reward aspect of the algorithm makes RLHF now a sort of multi-task optimization problem, and bring with the problem of how to weight the different categories. The discussion on the trade-off in (4.5) is interesting and helpful, but I think more could be said on how to pick a weighting for a general case where we just want a single “quality”. Also a very simple pareto analysis (probably trying more weightings of the reward) and some plots showing the final trade-off (ideally in terms of human reward?) would add more technical insight
>
> Task 1 experiments reflect the case where only a single “quality” is used, where “quality” means reward, and the fine-grained rewards are temporal (denser). The Pareto analysis would be feasible with Rouge scores, but the human evaluation is beyond the scope of what we can explore in this study and it would be particularly important for finding a good and general weight tradeoff.
>
>
> > ***Questions:***
>
> > The levels of granularity in the paper make sense for the three categories considered, but have you thought about the more general problem of allowing arbitrary granularities? It seems it is mostly a UX/feedback collection process, and that nothing in the algorithm would prevent it.
>
> Yes, our fine-grained RLHF framework can be applied to arbitrary granularities if the corresponding reward model is available. In fact, we collected our feedback data at the phrase level. However, we finally decided to use sub-sentence and sentence density levels, due to the fact that human annotators have low agreements on phrase boundaries (explained on line 196). Future work can explore more fine-grained feedback.
>
>
> > Is there plans to open-source the (fine-grained) RLHF code? Integrating this into popular packages for RLHF (like TRLX) could heavily boost the impact of this work
>
> Yes, we will add a link to our data and code in the final version.

---

> > ### Comment · Reviewer_uPU2 · 2023-08-14
> > **Acknowledgment of Response**
> >
> > I would like to thank the authors for their response.
> >
> > Overall I'm satisfied with responses.
> > I still think the paper would highly benefit from experiments with other models (even if not much larger than the current one), but think this is still a very valuable contribution, and will keep my review score.

---

### Official Review · Reviewer_Q8Ja · 2023-07-06

**Soundness:** 3 good
**Presentation:** 4 excellent
**Contribution:** 3 good
**Rating:** 7
**Confidence:** 4

**Summary:**

This paper studies how fine-grained RLHF improve over holistic preference RLHF.
The study is conducted on two tasks and fine-trained reward comes from sub-sentence, sentence, and full sequence level annotations on different feedback types (toxicity, relevancy, factuality and completeness).
Study show that fine-grained reward improved over holistic preference and demonstrated LM behaviors can be customized using combinations of feedback types.

**Strengths:**

- The paper is clearly written with good visual aid, has sufficient details on datasets, reward model training, descriptions on experiment hyperparameters and frameworks.
- The paper demonstrates benefits of fine-grained reward model on two tasks, with various reward density and reward types. It is convincing to see  it works across very different problem settings.
- There is clear and concise analysis on reward model and human rating agreement to understand the quality of reward model.
- Experiment sections contains relevant ablations that supports main results and analysis of how different types of reward model works together.

**Weaknesses:**

- (minor) in the detoxification analysis, I'm not fully convinced that the gain comes from reward, it is possible that the API has better accuracy annotating shorter sequence. Some additional analysis similar to your reward model human agreement analysis on task2 could help explain it better.
- I didn't find the mention of reward model size which I assume would be the same as the value model of t5-base which is smaller than the policy model. In previous works there is advantages of using larger than policy network value/reward function. It might be important to show what is the effect of reward/value model size on fine-grained reward vs holistic reward.
- (major) on task2, there lacks analysis comparing F.G.RLHF vs Pref.RLHF on holistic preference eval. It would be a lot more convincing if there is gain in that eval. Otherwise it could be argued that these different approaches are optimizing toward very different signals.
- the comparison ChatGPT using ROUGE score: it is a reference-based score that might not correlated well with human preference.  Using ROUGE score itself to compare these systems are not sufficient or even detrimental. Your eval with rel/fact/comp and the holistic preference are much better metrics.

**Questions:**

See weakness section.

---

> ### Author Rebuttal · Authors · 2023-08-09
>
> Thanks for your insightful comments! Please see our responses below.
>
> > ***1. (minor) in the detoxification analysis, I'm not fully convinced that the gain comes from reward, it is possible that the API has better accuracy annotating shorter sequence. Some additional analysis similar to your reward model human agreement analysis on task2 could help explain it better.***
>
> This is an interesting question. One possible analysis would require evaluating the API on hand-annotated data of different lengths, which is not available (to our knowledge). In our informal experiments, the toxicity is context-dependent and benefits from longer sections, which is why our sentence-level reward is calculated as API score difference of before and after the sentence generation (line 129).
>
>
> > ***2. I didn't find the mention of reward model size which I assume would be the same as the value model of t5-base which is smaller than the policy model. In previous works there is advantages of using larger than policy network value/reward function. It might be important to show what is the effect of reward/value model size on fine-grained reward vs holistic reward.***
>
> We follow Ouyang et al., 2022 to use reward models (Longformer-base) with a similar size as value models. It would be very interesting to explore the effect of reward/value model sizes on policy model performance, but this is out of the scope of our work. Some recent studies provide insights on the scaling of reward models: Scaling Laws for Reward Model Overoptimization; Training a Helpful and Harmless Assistant with Reinforcement Learning from Human Feedback. We will add this to our discussion section.
>
> > ***3. (major) on task2, there lacks analysis comparing F.G.RLHF vs Pref.RLHF on holistic preference eval. It would be a lot more convincing if there is gain in that eval. Otherwise it could be argued that these different approaches are optimizing toward very different signals.***
>
> **We do have the overall human preference results on fine-grained vs preference-based RLHF – 30.5% (win), 45% (tie), 24.5% (lose)**, where win/lose refers to fine-grained RLHF. We did not include the results because we believe the fine-grained evaluation would draw more insights and also due to space limits. We will try to squeeze these numbers in or put them in the Appendix.
>
> > ***4. the comparison ChatGPT using ROUGE score: it is a reference-based score that might not correlated well with human preference. Using ROUGE score itself to compare these systems are not sufficient or even detrimental. Your eval with rel/fact/comp and the holistic preference are much better metrics.***
>
> Due to the same concern with ROUGE, we conduct additional manual analysis (line 353) and provide qualitative examples (Appendix A) on ChatGPT results. We didn’t use our reward models to automatically evaluate ChatGPT outputs because reward models trained on T5-large do not generalize well to ChatGPT.  We will explain this more clearly in the paper.

---

> > ### Comment · Reviewer_Q8Ja · 2023-08-17
> > **acknowledge of response**
> >
> > Thank you authors for the response. I do not have further questions.

---

### Official Review · Reviewer_7ENw · 2023-07-07

**Soundness:** 4 excellent
**Presentation:** 4 excellent
**Contribution:** 3 good
**Rating:** 7
**Confidence:** 4

**Summary:**

This work improves the reinforcement learning from human feedback (RLHF) with more fine-grained reward signals.
For the feedback density, the authors suggest to reward the policy model at sub-sequence level.
The experiments on the detoxification task shows that the fine-grained rewards can outperform the holistic sequence level ones.
For the feedback diversity, they set three reward categories: 1) **rel.**, irrelevance, repetition, and incoherence, 2) **fact.**, incorrect or unverifiable facts, and 3) **comp.** information completeness.
The first two signals, i.e., rel. and fact., are with the sub-sentence and sentence level density, and the comp. is for the full sequence.
A new QA-FEEDBACK dataset with fine-grained error annotations and pair-wise preference ranks (for the baselien preference-based RLHF) is constucted for the training and evaluation.
Compared with the preference-based RLHF, the suggested fine-grained one behaves better on all three error types in both human and automatic evaluation.
This framework also demonstrates the promising ability of customizing LM behaviors by adjusting the weights of different reward models.

**Strengths:**

1. This work suggests a new perspective to RLHF for LM, where fine-grined and denser reward signals can benefit LMs.
2. The proposed fine-grained RLHF framework is simple and effective, validated via thorough experiments and analysis.
3. A new dataset is contributed for the research community, which is carefully created.
4. The paper is well-written and easy to follow.

**Weaknesses:**

1. The models used in this paper are relatively limited in size, only including large and base models. I'm curious how your method would scale to larger models, such as the more commonly used 13B and 30B models, and what kind of performance it would achieve.
2. LM customization is interesting, but lacks human evaluation. I suppose that incorporating human evaluation could help to further understand this.

**Questions:**

1. More different weights settings for the LM customization.
2. Efficiency and speed evaluations of the suggest method and baselines.

---

> ### Author Rebuttal · Authors · 2023-08-09
>
> Thanks for your insightful comments! Please see our responses below.
>
> > ***1. The models used in this paper are relatively limited in size, only including large and base models. I'm curious how your method would scale to larger models, such as the more commonly used 13B and 30B models, and what kind of performance it would achieve.***
>
> We agree that analyzing the scaling of our framework to larger models would bring insights to the community, but our work focuses on analyzing the effectiveness of using fine-grained reward for training LMs using RL, compared to holistic reward. Due to limited compute resources (RL training is expensive), we leave the exploration of model scaling for future work.  We do believe that, since many research groups and practitioners face similar computational budget issues, the findings we report on smaller models will be of wide interest.
>
> > ***2. LM customization is interesting, but lacks human evaluation. I suppose that incorporating human evaluation could help to further understand this.***
>
> We agree that human evaluation is important. Appendix A shows qualitative examples for LM customization. The anecdotal examples we’ve looked at support the findings (see line 301) from automatic evaluation results. We will expand the human assessment study in the appendix to have quantitative results.
>
> > ***3. More different weights settings for the LM customization.***
>
> We studied different weight settings in Table 5 as well. Table 5 provides results when removing each reward model (i.e., weight=0) on the valid set. Future work can include a more comprehensive personalization of the weights given the end task.
>
> > ***4. Efficiency and speed evaluations of the suggest method and baselines.***
>
> We use the same RL algorithm (PPO) as preference RLHF, and only observe **very small relative additional cost (~1% training time)** from running multiple reward models (not back-propagated), compared to preference RLHF during RL training. Specifically, the training time of each preference RLHF and fine-grained RLHF run takes 15.1 and 15.3 hours for long-form QA. The inference speed of preference and fine-grained RLHF is the same, roughly 0.7 seconds per example.

---

> > ### Comment · Reviewer_7ENw · 2023-08-19
> >
> > Thank you authors for the response. I do not have further questions.

---

### Official Review · Reviewer_tgte · 2023-07-07

**Soundness:** 4 excellent
**Presentation:** 3 good
**Contribution:** 2 fair
**Rating:** 6
**Confidence:** 4

**Summary:**

This paper introduces FINE-GRAINED RLHF, a framework that enables training and learning from reward functions that are fine-grained in different error categories. Through experiments on detoxification and long-form question answering, the authors demonstrate how learning with this FINE-GRAINED reward function leads to improved performance, supported by both automatic and human evaluation. The paper's contributions include a new framework for fine-grained reward learning, a new dataset for long-form question answering, and experiments demonstrating the effectiveness of the proposed approach. Though, the proposed method still has some limitations in terms of computational cost, algorithm complexity, and task-binding. In all, it has some contributions to the RLHF field.

**Strengths:**

- The paper introduces a new framework for fine-grained reward learning, which is a novel approach to language model training. The authors also provide a new dataset for long-form question answering, which can be used for future research in this area.
- The paper is well-written and clearly presents the proposed framework and experimental results. The authors provide detailed descriptions of their methodology and evaluation metrics, which enhances the quality of the paper.
- The proposed framework has the potential to improve the performance of language models by providing more fine-grained feedback during training.
- The idea is simple but reveals the potential of RLHF at finer grains.

**Weaknesses:**

- I know that data labeling is cumbersome and expensive, but the 3,853 training data collected in this work is still a bit small, not sure if this is enough to draw the conclusions mentioned in this work.
- The proposed method is a bit complicated, it will introduce more computational cost in RLHF, and it will introduce more hyper-parameters for fine-tuning.
- The design of T_i^k is not fully explored, which may have a big impact on the performance of the proposed method.

**Questions:**

- The authors say that "the higher the KL coefﬁcient, the lower the perplexity, and the higher toxicity". But only three KL coefficient is tested (i.e., 0.1, 0.2, 0.3) in the experiments, it is not enough to get such a conclusion. Also, what are the results on the generation fluency when the KL coefficient is 0, which means we do not employ a KL penalty? It would be better to explain the KL penalty in more detail.
- Why do you choose different models for different tasks? I.e., for the detoxiﬁcation experiments, the policy model is initialized with GPT2Large, and the value model is initialized with GPT2-base; for the long-form QA experiments, the policy model is initialized with a supervised ﬁne-tuned T5-large, and the value model is initialized with T5-base.
- Is the proposed method a good solution in terms of LLM's hallucination?

**Limitations:**

- The training dataset is not large enough (especially for the Long-Form QA task), which limits the verification of a conjecture: As the size of training data increases, the effect of the FG-RLHF proposed in this paper will weaken.
- The proposed method is expensive, not only in the additional computational cost of the reward model, but also in the different deﬁnitions of ﬁne-grained feedback in terms of the feedback types and the density level of each feedback type.

---

> ### Author Rebuttal · Authors · 2023-08-09
>
> Thanks for your insightful comments! Please see our responses below.
>
> > ***1. I know that data labeling is cumbersome and expensive, but the 3,853 training data collected in this work is still a bit small, not sure if this is enough to draw the conclusions mentioned in this work.***
>
> The amount of pairwise comparisons (6 comparisons per prompt) used here is **comparable to that used in other work**, e.g. WebGPT (20k comparisons). More importantly, we believe that it is an advantage to have a method that works well with a small amount of data.
>
> > ***2. The proposed method is a bit complicated, it will introduce more computational cost in RLHF, and it will introduce more hyper-parameters for fine-tuning***
>
> We use the same RL algorithm (PPO) as preference RLHF, and only observe **very small relative additional cost (~1% training time)** from running multiple reward models, compared to preference RLHF during RL training. Although it introduces more hyperparameters (reward model weights), they also bring the benefit of allowing LM training customization. We will add clarification on this to the next version of the paper.
>
> > ***3. The design of T_i^k is not fully explored, which may have a big impact on the performance of the proposed method.***
>
> The point of this work is to show that there are many options for rewards, including different types of rewards and different density levels. That space is bigger than can be fully explored in one paper. We hope the proof-of-concept offered in our work inspires more exploration with these and other tasks.
>
> > ***4. The authors say that "the higher the KL coefﬁcient, the lower the perplexity, and the higher toxicity". But only three KL coefficient is tested (i.e., 0.1, 0.2, 0.3) in the experiments, it is not enough to get such a conclusion. Also, what are the results on the generation fluency when the KL coefficient is 0, which means we do not employ a KL penalty? It would be better to explain the KL penalty in more detail.***
>
> Our observation **confirms findings from previous RLHF studies** like Ouyang et al., 2022 and Ramamurthy et al., 2023. As a result, we didn't expand more on it. For example, previous work provides extensive empirical evidence showing that the KL term is important for preventing reward hacking and this trend in particular. Therefore, we did not experiment with KL coefficient = 0. We will update the paper to make this point more clear.
>
>
> > ***5. Why do you choose different models for different tasks? I.e., for the detoxiﬁcation experiments, the policy model is initialized with GPT2Large, and the value model is initialized with GPT2-base; for the long-form QA experiments, the policy model is initialized with a supervised ﬁne-tuned T5-large, and the value model is initialized with T5-base.***
>
> ​​In order to have **fair comparisons with baseline models** for the two tasks, we chose GPT2-large and T5-large as the policy model for detoxification and long-form QA. In specific, prompts in REALTOXICITYPROMPTS are those known to elicit toxic generation from GPT2. T5-large is the current SOTA model and also the original baseline for ASQA. We simply use the same value model with the policy model (GPT2 or T5). However, due to GPU memory constraint, we use the base instead of large value models.  We will clarify the motivation for these choices in the paper.
>
>
> > ***6. Is the proposed method a good solution in terms of LLM's hallucination?***
>
> In Figure 3, we empirically show that fine-grained RLHF is more effective than preference-based RLHF in improving factual consistency with given evidence passages for the long-form QA task. We believe our framework has a great potential to help solve the general LLM hallucination issue, but believe further work is necessary to establish the scope of such benefits.
>
>
> > ***7. The training dataset is not large enough (especially for the Long-Form QA task), which limits the verification of a conjecture: As the size of training data increases, the effect of the FG-RLHF proposed in this paper will weaken.***
>
> Please see our response above regarding the training data size.
>
> > ***8. The proposed method is expensive, not only in the additional computational cost of the reward model, but also in the different deﬁnitions of ﬁne-grained feedback in terms of the feedback types and the density level of each feedback type.***
>
> Please see the above response about the computational cost of reward models. In the paper, we show that the fine-grained annotation task is easier for human annotators (more consistent in terms of agreements) because the definitions are more concrete, and annotation cost of fine-grained feedback is comparable to pairwise comparisons. We will add more discussion of this point in the next version of the paper.

---

### Author Rebuttal · Authors · 2023-08-09

We thank all the reviewers for your valuable feedback, and acknowledging our contributions in terms of novelty, effectiveness, and controllability of our fine-grained RLHF, as well as the data quality of QA-Feedback.

Please see our response to each reviewer’s comments separately below. We will add all corresponding clarifications and discussions to the next version of the paper. As we notice reviewers’ shared interest in our analysis on LM customization, we plan to expand the human assessment study on it to include more insights in the next version of the paper.

---

### Decision · Program_Chairs · 2023-09-21

**Decision:**

Accept (spotlight)

**Comment:**

The paper presents FINE-GRAINED RLHF, a framework that allows for training and learning from reward functions that are fine-grained in different error categories. The authors demonstrate improved performance on detoxification and long-form question answering tasks through learning with this FINE-GRAINED reward function.

The idea of  fine-grained reward learning is interesting and has  has the potential to improve language model performance by providing more detailed feedback during training. The framework demonstrates the promising ability of customizing LM behaviors by adjusting the weights of different reward models.

While there are some open questions about choice of base models, evaluation and generalization to other tasks, the paper presents a valuable contribution  and has potential for impacting future research in this area.